# Report

# AML patient blasts exhibit polarization defects upon interaction with bone marrow stromal cells

Khansa Saadallah [1,2], Benoît Vianay[1,2,3], Louise Bonnemay[1,2,3], Hélène Pasquer[4,5], Lois Kelly[4], Stéphanie Mathis[6], Cécile Culeux[4], Raphael Marie[4], Paul Arthur Meslin[4], Sofiane Fodil [6], Paul Chaintreuil[7], Emeline Kerreneur[7], Arnaud Jacquel [7], Emmanuel Raffoux [6], Rémy Nizard[8], Camille Lobry [4], Laurent Blanchoin [1,2,3], Lina Benajiba[4,5] & Manuel Théry [1,2,3 ✉]

## Abstract

**Hematopoietic stem and progenitor cells (HSPCs) polarize in contact with the bone marrow stromal cells constituting their niche. Given the role of cell polarity in protection against tumorigenesis and the importance of the niche in the progression of acute myeloid leukemias (AMLs), we investigated the polarization capacities of leukemic blasts. Using engineered micro-niches and centrosome position with respect to the contact site with stromal cells as a proxy for cell polarization, we show that AML cell lines and primary cells from AML patient blasts are unable to polarize in contact with healthy stromal cells. Exposure to AML patient-derived stromal cells compromises the polarization of healthy adult HSPCs and AML blasts from patients. When cultured in "bone-marrow-on-a-chip", stromal cells from a leukemic niche stimulate the migration of healthy HSPCs and AML blast. These results reveal the detrimental influences of both intrinsic transformation and extrinsic contact with transformed stromal cells on the polarization of AML blasts.**

**Keywords** Acute Myeloid Leukemia (AML); Hematopoietic Stem and Progenitor Cells (HSPCs); Artificial Niche; Microwell; Bone-marrow-on-a-Chip (BMoC)
**Subject Categories** Cancer; Haematology; Stem Cells & Regenerative Medicine

## Introduction

Acute myeloid leukemias (AMLs) constitute a diverse group of clonal hematopoietic disorders characterized by the accumulation of abnormal yet highly proliferative myeloid progenitor cells referred to as blasts (Moore et al, 1973; Agarwal et al, 2019). The sustained self-renewal and extensive proliferation of the blast population are fundamental aspects contributing to the persistence of leukemia (Griffin and Löwenberg, 1986). AMLs malignancies ultimately lead to failure of hematopoiesis by disrupting the normal commitment of hematopoietic stem cells (HSCs) and progenitor cells towards myeloid differentiation, resulting in cytopenia and debilitating complications contributing to the mortality associated with AMLs (Miraki-Moud et al, 2013). The genesis of this hematological malignancy stems from a confluence of environmental and genetic modifications. It is widely recognized as a clinically heterogeneous disease with significant variability in post-treatment survival influenced by factors such as age, blast cell morphology, cytogenetic abnormalities, and gene mutations (DiNardo and Cortes, 2016).

There is speculation that leukemic cells hijack and disrupt the supportive microenvironment of HSCs through several potential targets exploited by leukemic stem cells (LSCs), including adhesion molecules (CD44) or chemokine receptors (CXCR4) (Konopleva and Jordan, 2011). This phenomenon has the potential to shift the balance of microenvironments from supporting steady-state hematopoiesis to conditions favoring accelerated expansion of leukemic cells, potentially contributing to leukemogenesis (Kode et al, 2014) and chemoresistance development. The characterization of both healthy and diseased hematopoietic cells is inextricably intertwined with the dynamic compartment in which they develop, the bone marrow niche. Consisting of stromal (mesenchymal stem cells (MSCs), osteoblasts, endothelial cells (ECs)…) together with hematopoietic cellular components, the hematopoietic niche exerts a substantial influence on hematopoiesis, the development, and/or maintenance of leukemia, not to mention the alteration of the niche itself in the context of leukemic transformation (Duarte et al, 2018). It is therefore imperative to understand the fundamental role of the microenvironment in the progression of the disease to gain a

[1]CytoMorpho Lab, Human Immunology Pathophysiology Immunotherapy (HIPI), U976, Institut de Recherche Saint Louis (IRSL), Hôpital Saint Louis, Université Paris Cité, Paris, France. [2]CytoMorpho Lab, Chimie Biologie Innovation (CBI), UMR 8231, Ecole Supérieure de Physique et Chimie Industrielle de la Ville de Paris (ESPCI), Institute Pierre-Gilles de Gennes, Paris, France. [3]CytoMorpho Lab, Laboratory of Cellular and Vegetal Physiology, UMR 5168, CEA, INRA, CNRS, University of Grenoble-Alpes, Interdisciplinary Research Institute of Grenoble, Grenoble, France. [4]INSERM U944, Institut de Recherche Saint Louis (IRSL), Université Paris Cité, Paris, France. [5]Centre d'Investigation Clinique INSERM 1427, Hôpital Saint Louis, Assistance Publique-Hôpitaux de Paris (APHP), Université Paris Cité, Paris, France. [6]Department of Hematology and Immunology, Hôpital Saint Louis, Assistance Publique-Hôpitaux de Paris (APHP), Université Paris Cité, Paris, France. [7]INSERM Unit 1065, Mediterranean Center of Molecular Medicine (C3M), University of Côte d'Azur, Nice, France. [8]Service de chirurgie orthopédique et traumatologique, Hôpital Lariboisière, Assistance Publique-Hôpitaux de Paris (AP-HP), Université Paris Cité, Paris, France. ✉E-mail: manuel.thery@cea.fr

comprehensive understanding of the intrinsic communication between all components within the hematopoietic niche. Unfortunately, the complex interplay of chemical and physical factors in the niche makes it challenging to discern the specific contributions of stromal and hematopoietic cells to the development of AML.

In various tissues, maintaining the balance between stem cell expansion and differentiation relies on the regulation of cell polarity and asymmetric cell divisions (Marciniak-Czochra et al, 2009; Wodarz and Näthke, 2007; Martin-Belmonte and Perez-Moreno, 2012; Prado-Mantilla and Lechler, 2023). There is growing evidence to support the tumor-suppressive roles of polarity proteins in mammals, in particular their impact on cell-to-cell or cell-to-matrix interactions (Lee and Vasioukhin, 2008). HSCs located at the apex of normal hematopoiesis, undergo polarization during migration, involving shape alterations and protein segregation (Freund et al, 2006). It has been elucidated that intercellular contact led to HSC polarization by inducing the formation of a cellular protrusion called magnupodium (Francis et al, 1998). This protrusion formation occurs alongside the polarization of certain proteins, such as CD133 (Prominin-1) (Bidlingmaier et al, 2008) and CD44 (Wagner et al, 2008). Consistent with in vivo observations (Coutu et al, 2017), our laboratory has recently found that HSCs and HSPCs manifest a variety of shapes, with a predominant elongated morphology observed during interactions with osteoblasts or ECs (Bessy et al, 2021). HSPC appeared capable of polarizing in contact with stromal cells, as revealed by the position of their centrosomes toward the magnupodium they formed in contact with stromal cells. This opens the question of the ability of leukemic blasts to polarize in contact with stromal cells and the role of this polarization in the maintenance of proper hematopoiesis or AML development.

Although the influence of polarity on hematopoietic stem cell function is a relatively new field, it is gaining traction, especially in the context of hematopoietic malignancies such as leukemias. The relevance of polarity in disease initiation, progression, or suppression, is intricately linked to an understanding of leukemia's origin. For instance, inhibition of Llgl1, a component of the Scribble complex known for its role in HSCs renewal, diminishes survival in AMLs (Mohr et al, 2018). The inactivation of Scribble partners induces dysregulation in the proliferation, signaling, and motility of HSCs. Also, the disruption of Yap1/Taz co-polarization with Cdc42 through the Scribble complex precipitates the loss of quiescence and self-renewal in HSCs (Althoff et al, 2020). Furthermore, there is evidence indicating a loss of polarity in HSCs during the aging process, characterized by depolarization of Cdc42 (Florian et al, 2012). The restoration of polarity through Cdc42 depletion facilitates the differentiation of AML cells (Mizukawa et al, 2017). Conversely, polarity alterations can prompt the redistribution of pro-differentiation cues, consequently suppressing tumorigenesis. Therefore, inhibition of Lis-1 enhances the asymmetric inheritance of Numb, thereby fostering the differentiation of AML cells (Zimdahl et al, 2014). These observations underscore the importance of investigating polarity, as it is intricately linked to mechanisms compromised in the leukemic context.

This study employs engineered artificial niches to investigate the communication between AML blasts and bone marrow stromal cells. We used cell lines and primary cells collected from healthy donor and AML patients. A first method employs cell-sized microwells constraining a pair of a stromal cell with a

hematopoietic cell, allowing prolonged cell–cell interactions while restricting migration. A second method involves a microfluidic bone-marrow-on-a-chip (BMoC) for three-dimensional coculture, allowing both migration and interaction in adjacent endothelial and endosteal compartments.

# Results and discussion

To investigate the functional impact of hematopoietic cells and stromal cell interactions, polyacrylamide microwells coated with fibronectin were employed to isolate these cells for a 24-h interaction period, followed by fixation and immunofluorescence (Fig. 1A) (Bessy et al, 2021). Centrosome positioning was utilized as a polarity marker, and the quantitative analysis of its localization regarding the contact region with stromal cells served as a measurement of the cell polarization index (Fig. 1A) (Bessy et al, 2021).

Pairs of osteoblasts (hFOBs) and cord blood-derived (CB) CD34$^+$HSPCs were confined in microwells, and their interactions were compared with those involving AML leukemic cells from MOLM-14 and NOMO-1 cell lines (Fig. 1B). Notably, CB HSPCs demonstrated significant polarization (Fig. 1C) with a distinctive magnupodia pseudopod formation upon attachment to osteoblasts aligning their centrosome toward the contact zone (Fig. 1B). Conversely, both MOLM-14 and NOMO-1 blasts predominantly exhibited a symmetrical, rounded morphology (Fig. 1B) with random polarization during their interactions with osteoblasts (Fig. 1C). However, cell lines have acquired specific characteristics associated with their adaptation to culture conditions, potentially deviating from the behavior exhibited by blasts within the bone marrow. The prolonged absence of stromal cells in their environment may compromise their anchorage machinery. Alternatively, defective anchorage and polarization mechanisms could represent a distinctive feature of AML progression. To further investigate these possibilities, it was necessary to work with primary cells derived directly from patients.

Primary cells, specifically bone marrow-derived MSCs (MSCs) and bone marrow-derived CD34$^+$-HSPCs, were sourced from healthy donors (HD) undergoing hip surgery for prosthesis placement. For the malignant counterpart, primary bone marrow-derived MSCs (AML MSCs) and blasts were harvested from AML patients. AML cells were obtained from individuals with distinct mutational statuses (Table 1) displaying high percentages of blasts from patients with an age range of ~50 years (aged between 20 and 70), facilitating a meaningful comparison between them while mitigating the influence of age-related biases (Table 2). The cell collection was performed at the time of diagnosis, prior to any treatment or chemotherapy. Thus, freshly collected AML patient blasts and healthy HSPCs were promptly plated into the engineered niches alongside stromal cells. These cells were identified using an advanced cytometry analysis strategy (Plesa et al, 2024), incorporating LSC gating based on CD34, CD38, CD45, and CD117, supplemented with markers CD90Thy-1, Mix (CD97/CLL1/TIM3), CD45RA, and CD123 (Table 3; Figs. EV1 and EV2).

The interaction dynamics between adult HSPCs and osteoblasts (from osteoblastic cell line) (Fig. 2A) or HD MSCs (primary cells) (Fig. 2B) in microwells unveiled their capability of magnupodium formation directed toward the contact zone. Notably, blasts from three AML patients (Pat#1-3) mainly exhibited rounded shapes

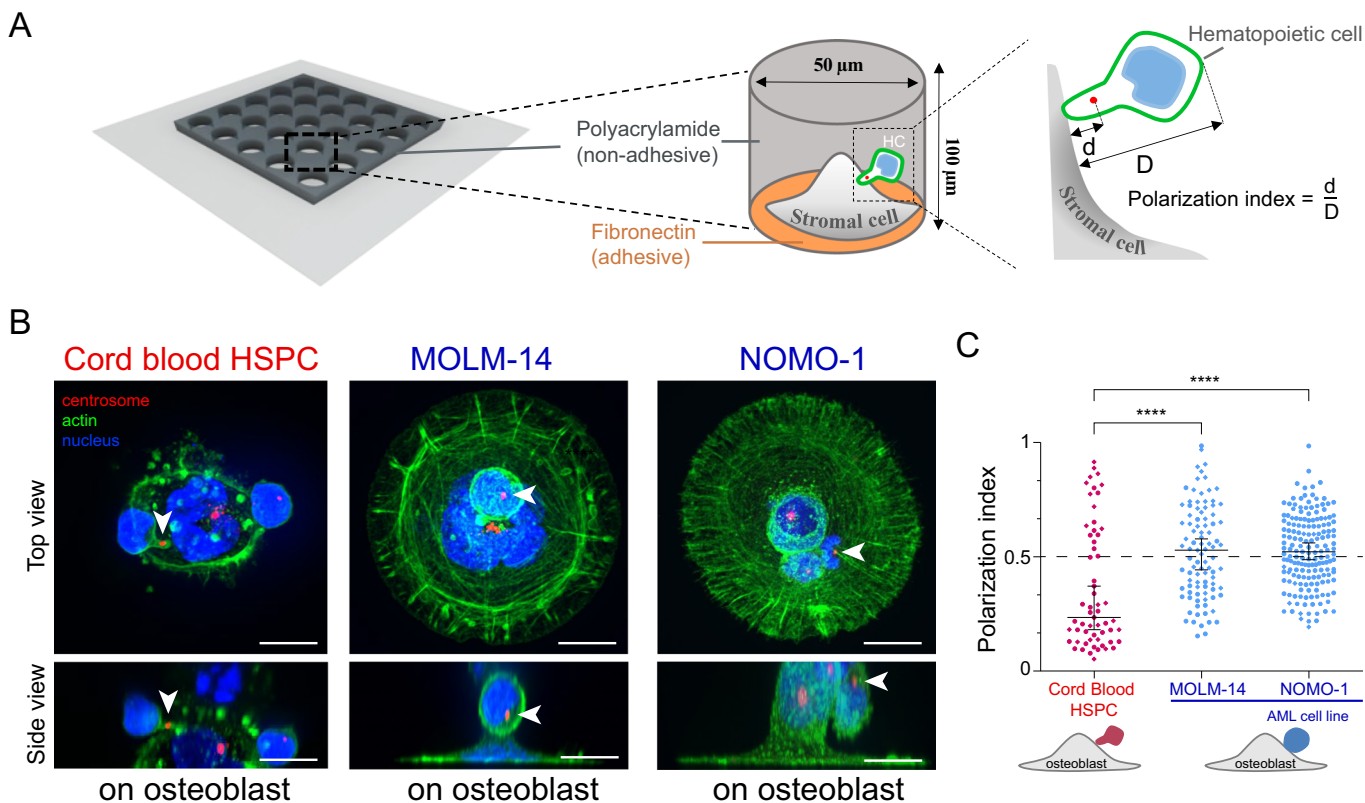

**Figure 1. AML cell lines failed to polarize when interacting with osteoblasts within microwells.**

(A) Schematic drawing representing microwells on a cover slide. The first zoom illustrates the non-adhesive edges (polyacrylamide) and the adhesive bottom (fibronectin coating), allowing the attachment and confining of stromal cells. The second magnification illustrates the position of the centrosome in the hematopoietic cell relative to cell contact with the stromal cell and the measurement of the polarization index (d/D). (B) Immunostainings of the centrosome (red), the nucleus (blue) and the actin cytoskeleton (green) in Cord blood (CB) HSPC and AML cell lines interacting with osteoblasts (hFOB) within microwells. Images were acquired in 3D. The top row shows a top projected view and scale bars correspond to 10 μm. Bottom row shows a side projected view and scale bars correspond to 10 μm. CB HSPCs form a magnupodium in contact with stromal cells. MOLM-14 and NOMO-1 leukemic blasts are both round and do not exhibit a magnupodium upon interaction with the osteoblast. (C) Graphs show the quantification of the polarization index of AML cell lines, MOLM-14 and NOMO-1, in contact with osteoblasts (hFOB) in comparison to CB HSPCs within microwells. Experiments were repeated two or three times and data were pooled ($n_{CB\ HSPCs} = 59$, $n_{MOLM-14} = 95$; $n_{NOMO-1} = 173$). Each spot corresponds to the measurement of the polarity of a single cell. The various shapes of the spots correspond to distinct experiments. Black bars show the median value and the standard deviation of the distribution. Differences between populations were evaluated using non-parametric test Kruskal–Wallis ANOVA with a P value < 0.0001 (****).

**Table 1. AML patient clinical features.**

| Patient | BM blasts% | LSCs% | Extracted cells | Karyotype | Mutational status |
|---|---|---|---|---|---|
| 1 | 67 | 1.52 | Blasts | Normal | IDH2; DNMT3A; BCOR |
| 2 | 98 | 0.98 | Blasts | Normal | NPMI; DNMT3A; RAS/MEK/ERK signaling |
| 3 | 92 | 0 | Blasts | Normal | ITD; CEBPA |
| 4 | 52 | 0.02 | MSCs | 45,X,-Y,t(8;21)(q22;q22)[22] | ITD-high; CBFβ-MYH11; MGA |
| 5 | 60 | 0 | MSCs & Blasts | 47,XY, inv(16)(p13q24),+21[20] | CBFβ-MYH11 |
| 6 | 82 | 0 | MSCs | 46,XY[20] | NRAS; PTPN11; Ras/MEK/ERK signaling; CEBPA; GATA2 |
| 7 | 98 | 1 | Blasts | Normal | ITD-high; RUNX1 |
| 8 | 35 | 0.01 | Blasts | 46,XX,t(11;19)(q23;p13)[20] | NRAS; KRAS; Ras/MEK/ERK signaling |
| 9 | 54 | 0 | MSCs | 46,XX[20] | TET2; DNMT3A; RUNX1; ASXL2; BCOR |
| 10 | 81 | 32 | LSCs/Blasts | 46,XY,-3,-5,del(7)(q22),-9,-11,del(17)(p11),-17[3], + 5-6mar[cp5]/46,XY[2] | U2AF1; TP53 |

Anonymized clinical data of AML patient (1–10) donors of either AML MSCs or blasts, bone marrow (BM) blast percentage, leukemic stem cell (LSC) percentage, nature of the extracted cells from the bone marrow sample, patient karyotype, and mutated genes are presented.

**Table 2.** AML patient cytogenetic profiles.

| Patient | Age | Gender | Disease status | ELN Classification 2022 | Time sampling |
|---|---|---|---|---|---|
| 1 | 49 | Male | De novo AML | Intermediate | patient died |
| 2 | 57 | Female | De novo AML | Intermediate | Complete remission |
| 3 | 44 | Male | De novo AML | Favorable | Primary refractory |
| 4 | 27 | Male | De novo AML | Favorable | Complete remission |
| 5 | 20 | Male | De novo AML | Favorable | Complete remission |
| 6 | 21 | Male | De novo AML | Intermediate | Complete remission |
| 7 | 70 | Male | De novo AML | Adverse | Complete remission |
| 8 | 59 | Female | De novo AML | Adverse | Not available |
| 9 | 65 | Female | De novo AML | Adverse | patient died |
| 10 | 68 | Female | De novo AML | Adverse | patient died |

Ages, genders, disease statuses, AML classification (ELN classification 2022), and time sampling of anonymized AML patient donors of hematopoietic cells or stromal cells are displayed.

**Table 3.** AML patient hematopoietic cell immunophenotypic profiles (blasts and LSCs).

| Patient | CD45⁺dim blastic population phenotype | | LSC population phenotype |
|---|---|---|---|
| 1 | First population: CD34⁺partial CD38⁺CD117⁻HLA-DR⁻ CD123⁺ partial cCD13⁺ CD13⁺ CD33⁺ CD90- CLL1/TIM3/CD97⁺ CD45RA⁺. | Second population: CD34⁻ CD38⁺CD117⁻HLA-DR⁺dim (30%) CD123⁺dim cCD13⁺CD13⁺ CD33⁺ CD90⁻ CLL1/TIM3/CD97⁺CD45RA⁺, expressing monocytic markers CD4⁺CD11b⁺CD14/CD64. | 1.52% LSCs: CD34⁺ CD38⁻ CD117⁻ CD13⁺ CD33⁺ CD90⁻ CD123⁺ CD45RA⁺ CLL1/TIM3/CD97⁺. |
| 2 | CD34⁻ CD38⁺ CD117⁺ CD123⁺ cCD13⁺ CD13⁺partial CD33⁺ CD65⁺ CD90⁻ CD45RA⁺ CLL1/TIM3/CD97⁺ | | 0.98% LSCs: CD34⁺ CD38- CD117⁺ CD13⁻ CD33⁺ CD90⁻ CD45RA⁺ CLL1/TIM3/CD97⁺. |
| 3 | cCD13⁺/CD13⁺partial CD33⁺partial CD34⁺ CD38⁺ CD117⁺ CD64⁺partial CD7⁺ CD123⁺partial CD45RA⁺partial CD97/CLL1/TIM3⁺ CD90⁻. | | No detectable LSCs at a sensitivity threshold of 10⁻⁴ |
| 5 | First population: CD45dim CD34⁺ CD117⁺ CD13⁺ CD33⁺partial HLA-DR⁺dim CD36⁻ | Second population: CD34⁺partial CD117⁺ CD13⁺ CD33⁺ HLA-DR⁺ Expressing monocytic markers CD64⁺ CD36⁺ CD11b⁺ CD4⁺ CD14⁻ | No detectable LSCs at a sensitivity threshold of 10⁻⁴ |
| 7 | CD34⁻ CD38⁺partial CD13⁺ CD117⁺partial CD33⁺ HLA-DR⁺ partial (30%) CD123⁺ | | 1% LSCs: CD34⁺ CD38⁻ CD7⁺partial CD33⁺ CD117⁺partial CD13⁺ CD123⁺ CD97/CLL1/TIM3⁺ CD45RA⁺ |
| 8 | CD34⁻ CD117⁻ CD13⁺partial CD33⁺ CD38⁺ HLA-DR⁻ CD123⁺partial CD97/TIM3/CLL1⁺ CD45RA⁺partial with expression of monocytic markers CD64⁺ CD16⁺partial CD4⁺dim CD14⁺partial | | 0.01% LSCs: CD34⁺CD38-CD117⁺partial CD13⁺partial CD33⁺partial CD123⁺partial CD45RA⁺partial CD97/CLL1/TIM3⁻ |
| 10 | CD34⁺ CD38⁺dim/⁻ CD117⁻ CD13⁺ CD33⁺ HLA-DR⁺ partial (89%) CD123⁺partial CD45RA⁻ CD97/CLL1/TIM3⁺partial | | 32% LSCs: CD34⁺ CD38⁻ CD13⁺ CD33⁺ CD7⁺partial CD123⁺partial CD97/CLL1/TIM3⁺ CD45RA⁻ |

Flow cytometry analysis of anonymized AML patient cells highlighting the predominant blastic and/or LSC population fractions within the bone marrow samples. Only immunophenotypic profiles of hematopoietic cell donor patients are presented.

when in contact with the healthy stromal cell, devoid of the distinctive presentation of a narrow pseudopod observed in adult HSPCs (Fig. 2B).

Furthermore, in a manner analogous to the description of cord blood-derived HSPCs (CB HSPCs), adult CD34⁺-HSPCs from healthy donors exhibited a preserved ability to orient their centrosome towards osteoblast (Fig. 2C) and healthy bone marrow stromal cell (Fig. 2D). In contrast, the three AML blasts from patient donors displayed a random distribution of their centrosome localization in all conditions (Fig. 2C,D). Consistent with the pattern observed in blasts, AML patient cells enriched with LSCs (Patient #10) also exhibited a stochastic distribution of centrosomes (Fig. 2C,D). Thus, the polarization mechanism

appeared functional in healthy adult HSPC and intrinsically perturbed in AML blast. However, the polarization process is contingent upon the interplay between diverse cell types, so it might also be regulated by the stromal cells. The AML niche being instrumental to tumoral development, it appeared necessary to specifically test the potential role of stromal cells from AML patients on the polarization of healthy HSPC and leukemic blasts.

To that end, we first tested the ability of HSPCs derived from healthy donors (HD HSPCs) to polarize in contact with MSCs derived from healthy donors or AML patients (Fig. 3A). Interestingly, HD HSPCs from the same donor that polarized in contact with HD MSCs, displayed a random orientation of their

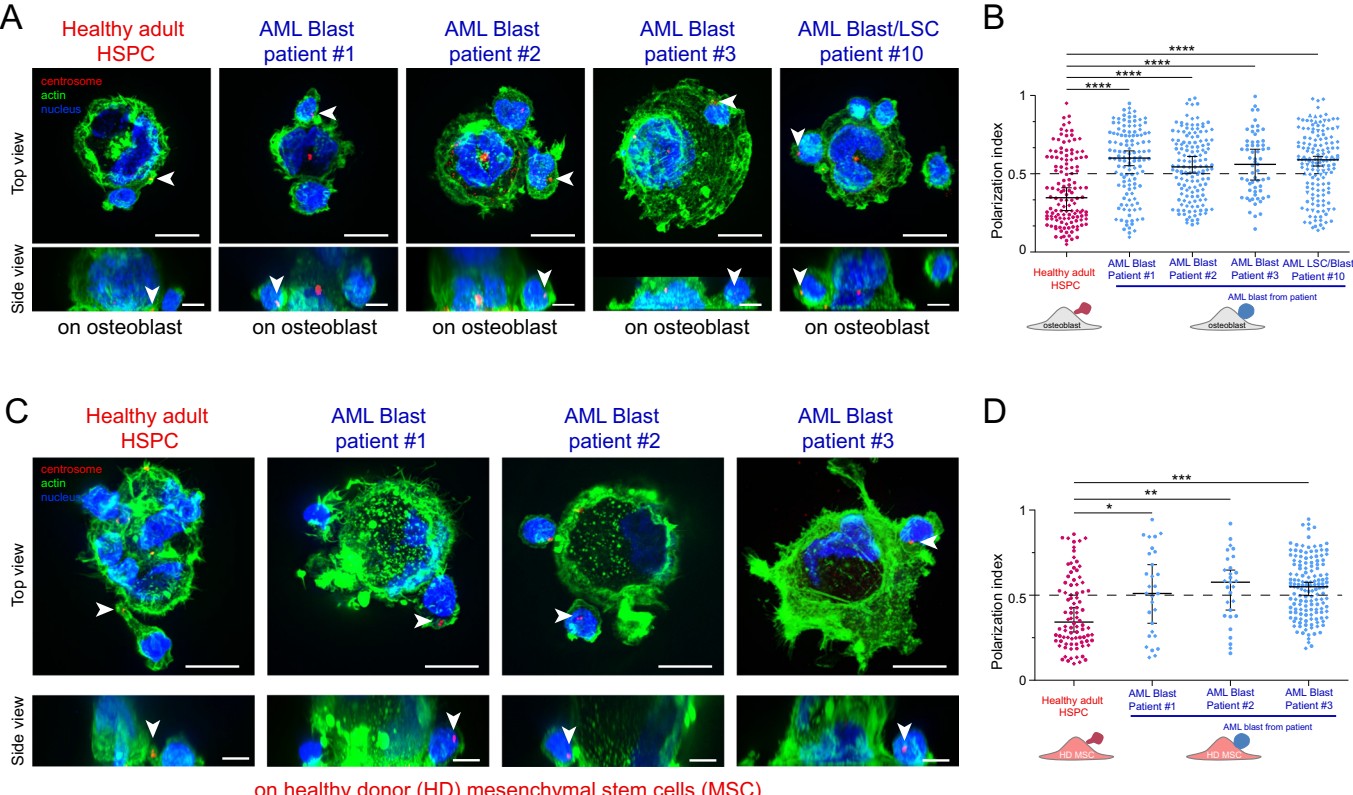

**Figure 2. AML patient blasts failed to polarize when interacting with various stromal cells from the bone marrow.**

(A) Immunostainings of the centrosome (red, pointed at with white arrowheads), the nucleus (blue) and the actin cytoskeleton (green) in HSPCs from healthy adult donor and AML primary leukemic cells from patients #1, #2, #3, and #10 interacting with osteoblasts (hFOB) in microwells. Images were acquired in 3D. The top row shows a top projected view, and scale bars correspond to 10 µm. Bottom row shows a side projected view and scale bars correspond to 5 µm. (B) Same as (A) with adult HSPCs and AML primary leukemic cells from patients #1, #2, and #3 interacting with healthy donor MSCs (HD MSCs) in microwells. Similar to CB HSPCs, adult HSPCs elongate and form magnupodia where can be located their centrosomes in close vicinity to the contact site with stromal cells (osteoblast or HD MSC). AML primary blasts from patients #1, #2, and #3 and AML primary LSCs from patient #10 exhibit large zones of contact with no sign of magnupodia in contact with the stromal cell (osteoblast or HD MSC). (C) Quantification of the polarization index of AML patient blasts (blue dots), compared to HSPCs from healthy adult donor (red dots), in contact with osteoblasts (hFOB) within microwells. Experiments were repeated three times and data were pooled ($n_{adult\ HSPCs}$ = 132, $n_{patient\ 1\ blasts}$ = 123; $n_{patient\ 2\ blasts}$ = 142, $n_{patient\ 3\ blasts}$ = 61; $n_{patient\ 10\ LSCs}$ = 149). Each spot corresponds to the measurement of the polarity of a single cell. The various shapes of the spots correspond to distinct experiments. Black bars show the median value and the standard deviation of the distribution. Differences between populations were evaluated using a Kruskal–Wallis ANOVA test with *P* values < 0.0001 (****). (D) Same as (C) for AML patient blasts (blue dots) and HSPCs from healthy adult donor (red dots) in contact with healthy donor (HD) mesenchymal stem cells (MSCs) within microwells. Experiments were repeated three times and data were pooled ($n_{adult\ HSPCs}$ = 88, $n_{patient\ 1\ blasts}$ = 148; $n_{patient\ 2\ blasts}$ = 29, $n_{patient\ 3\ blasts}$ = 29). Differences between populations were evaluated using a Kruskal–Wallis ANOVA test with *P* values of 0.03, 0.004 and <0.0001 (*, ** and ****). AML primary blasts from patients #1, #2, #3 and LSCs from patient #10 display a random distribution of their centrosome positioning while interacting with osteoblasts and HD MSC, whereas adult HSPC from HD were polarized in contact with both cell types.

centrosomes in contact with AML MSCs (Fig. 3B). This observation demonstrated that transformed stromal cells could impair the polarization of healthy HSPC, although the machinery underlying the control of centrosome position was functioning properly in these cells. Furthermore, when mimicking the conditions within the bone marrow of patients, i.e., by seeding patient-derived blasts onto AML-derived stromal cells, we observed that the AML patient blasts not only lacked the ability to polarize in contact with AML MSCs but even exhibited a tendency to polarize in the opposite direction (Fig. 3B). The majority of centrosomes were positioned toward the uropod, situated at the rear of the cell, as shown by the values of polarity indices between 0.5 and 1 (Fig. 3B). This orientation of polarization toward the rear of the blasts was reminiscent of the polarization observed in migrating leukocytes (Sánchez-Madrid and Serrador, 2009; Kopf and Kiermaier, 2021).

This suggests that the observed inverted polarization of AML blast on AML-derived stromal cells may result not only from defective anchorage but also from the stimulation of their migration properties.

We took advantage of our recent design of a microfluidic chip that reconstitutes a simplified but transparent bone marrow in a microfluidic circuit, i.e., a "bone-marrow-on-a-chip" (BMoC), in order to investigate cell migration properties (Souquet et al, 2021; Bessy et al, 2021). Our BMoC incorporates distinct compartments for stromal cells, supporting long-term culture of osteoblasts, endothelial cells, as well as primary stromal cells from healthy donors or AML patients, all encapsulated in a collagen-fibrin-based hydrogel (Fig. 4A). The three-dimensional environment provided by this hydrogel allows hematopoietic cells to visit all compartment and possibly establish contact, detach, or migrate in contact with

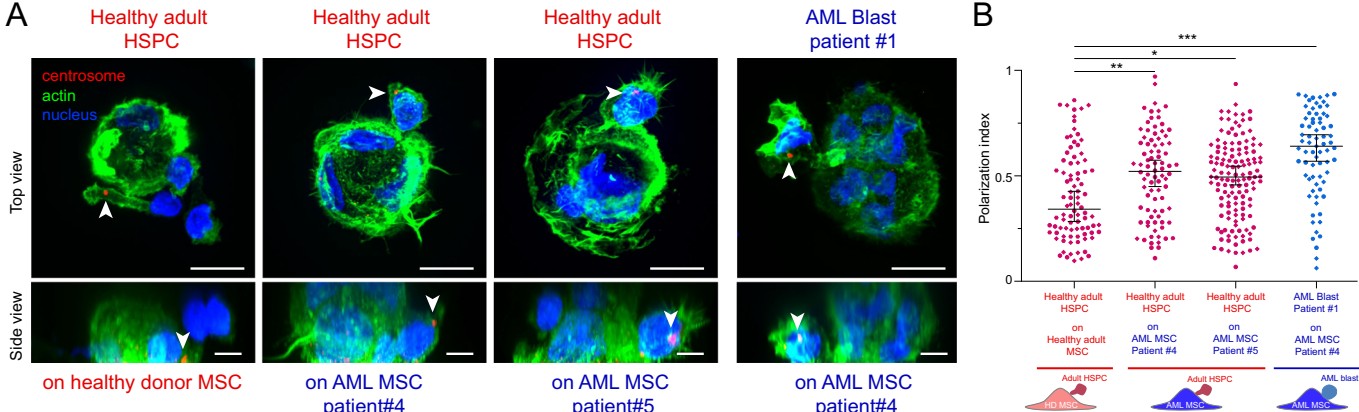

**Figure 3. Contact with stromal cells from AML patients impairs the polarization of both adult HSPCs from healthy donors and leukemic blasts from AML patients.**

(A) Immunostainings of the centrosome (red, pointed at with white arrowheads), the nucleus (blue) and the actin cytoskeleton (green) of adult HSPCs from healthy donor (HD) interacting with either healthy or AML bone marrow stromal cells. Images were acquired in 3D. Top row shows a top projected view and scale bars correspond to 10 μm. Bottom row shows a side projected view, and scale bars correspond to 5 μm. (B) Quantification of the polarization index of adult HSPCs from HD (red dots) and AML patient blasts (blue dots) upon interaction with AML MSCs in the microwells. Experiments were repeated three times and data were pooled ($n_{\text{adult HSPC / HD MSC}} = 88$, $n_{\text{adult HSPC / Patient 4 MSC}} = 84$; $n_{\text{adult HSPC / Patient 5 MSC}} = 135$, $n_{\text{Patient 1 blasts / Patient 4 MSC}} = 73$). Each spot corresponds to the measurement of the polarity of a single cell. The various shapes of the spots correspond to distinct experiments. Black bars show the median value and the standard deviation of the distribution. Differences between populations were evaluated using a Kruskal–Wallis ANOVA test with $P$ values of 0.0060, 0.0222, and <0.0001 (**, *, ****). In adult HSPCs from HD interacting with AML MSCs derived from patient #5 and patient #4 the centrosome is mispositioned. In AML patient blasts interacting with AML MSCs from patient #4, the centrosome is mispositioned. Furthermore, AML patient blasts display a reverted polarization (polarity index >0.5) in contact with AML MSCs.

specific stromal cells. The BMoC design also enables continuous monitoring of live cells, as well as fixation, immunostaining, and high-resolution imaging of fixed samples (Fig. 4A).

We first tested the ability of HSPCs to polarize in contact with either endosteal (hFOBs) (Fig. 4B,C) or endothelial (HUVECs) cells (Fig. 4D,E) in the BMoC. In accordance with our previous observations of confined hematopoietic-stromal cell doublets in microwells, HSPCs appeared capable of polarizing towards both cell types, while AML cell lines (MOLM-14 and NOMO-1) exhibited a random polarity (Fig. 4C,E). We then investigated the migration of AML cell lines and patient blasts in compartments containing stromal cells from healthy donors. Remarkably, the cell line exhibited minimal movement, whereas patient blasts demonstrated faster and more processive migration tracks (Fig. EV3). This observation suggested that long-term culture of cell lines in flasks without physiological anchorage to a physiological niche may have led to a progressive loss of migration properties. This prompted us to shift our focus to primary hematopoietic cells.

We analyzed the migration trajectories of healthy donor-derived HSPC (HD HSPCs) and AML-derived blasts in contact with stromal cells from healthy or AML patients (Fig. 5A). Interestingly, HD- HSPCs exhibited faster and more persistent movements when in contact with AML-derived stromal cells compared to stromal cells derived from healthy donors (Fig. 5B). These findings align with our observation in microwells, indicating defective polarization and thus increased ability of healthy HSPCs to detach and move away when in contact with stromal cells derived from AML patients. Furthermore, upon reconstituting the conditions mimicking AML patients (blasts and MSC derived from AML patients), we found that blasts, displayed even more accelerated and extensive movement on AML-derived stromal cells than on healthy stromal cells (Fig. 5B). Interestingly, these data showed that the leukemic

niche actively promotes the migration of hematopoietic cells, whether they are transformed or not. But they also demonstrated that the combination of leukemic blast and leukemic niche leads to higher motility than any other combination of hematopoietic and stromal cells.

In this study, we identified the impaired ability of blasts to engage in interaction and polarization when in contact with the stromal cells of their niche. Initially, we reported the polarization of healthy cord blood-derived HSPCs in contact with osteoblastic or endothelial cell lines (Bessy et al, 2021). Here, we confirmed that adult HSPCs have retained this property. Interestingly, we found that adult HSPCs could not polarize in contact with leukemic stromal cells. This suggests that the leukemic niche can have a detrimental impact on the structural organization of healthy HSPCs. Conversely, we found that leukemic blasts could not polarize toward either healthy or leukemic stromal cells. So, collectively, our findings demonstrated that the malignancy of either stromal or hematopoietic cells from AML patients hinders polarization of hematopoietic cells. Moreover, they indicated that the combination of a leukemic blast in a leukemic niche fully reverses blast polarity and promotes its migration. This showed that both intrinsic and extrinsic factors are involved in the regulation of the polarization of hematopoietic progenitors, as it has been demonstrated in other tissues (Andrews et al, 2022). Furthermore, our data showed that both hematopoietic cells and stromal cells from the niche contribute to the morphological transformation of leukemic cells, contributing to the substantial body of evidence indicating the critical role of this interplay in leukemia progression (Duarte et al, 2018).

Further investigation is warranted to elucidate the molecular pathways that are affected by the leukemic context and lead to the loss of cell polarity. Our work showed the implication of the SDF1-

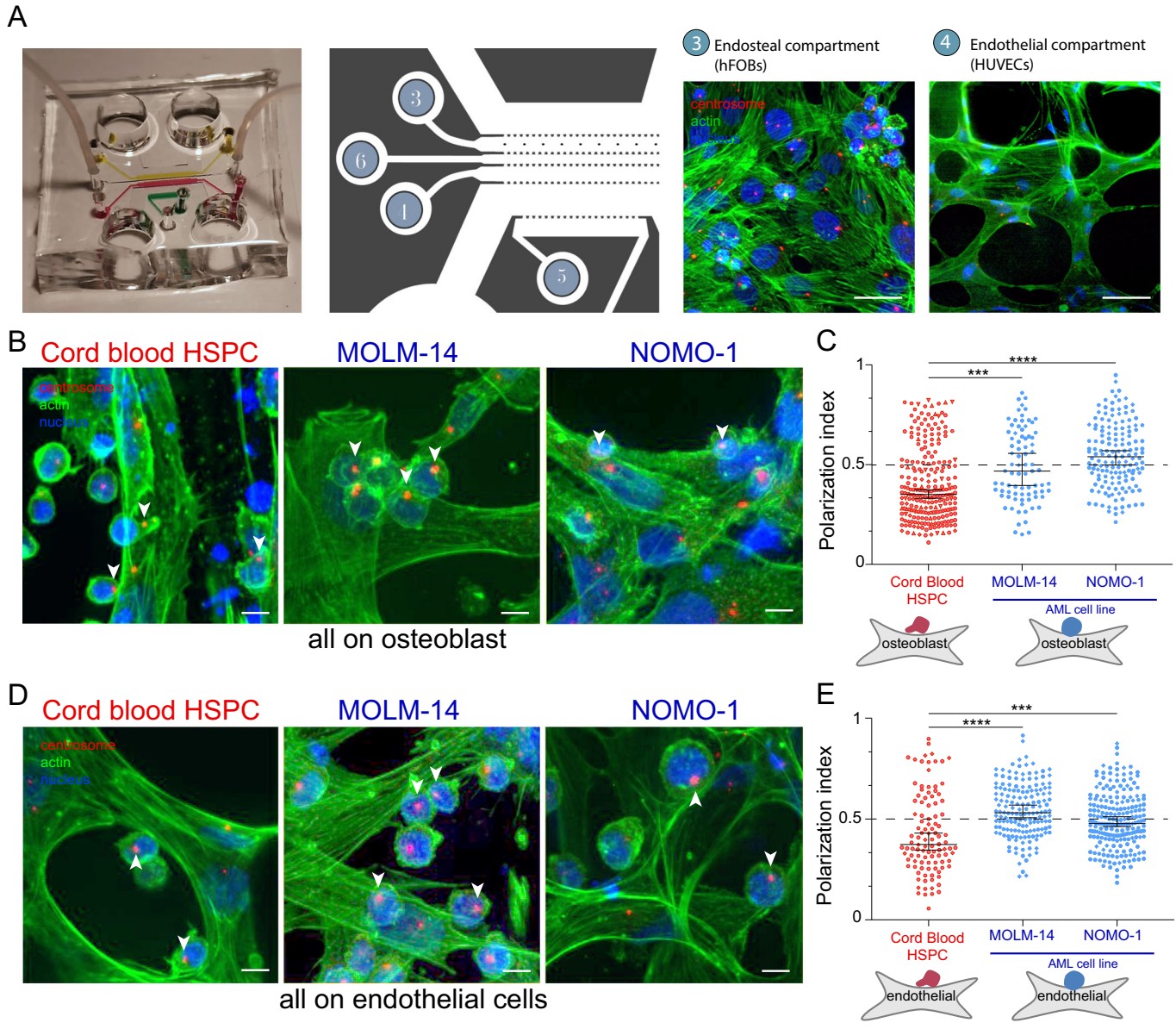

**Figure 4. AML cell lines and AML patient blast failed to polarize when cultured with stromal cells in a bone marrow-on-a-chip (BMoC).**

(**A**) The microfluidic circuit of the BMOC comprises 6 parallel channels. Channels (1) and (2) (not shown) are committed to medium supplementation. Channels (3), (4) and (5) are dedicated to endosteal (hFOBs), endothelial (HUVECs) and fibroblastic cells (NHLFs), respectively. All stromal cells are cultured in an hydrogel composed of collagen and fibrin. Endosteal compartment harbors central pillars to resist hydrogel contraction due to osteoblasts traction forces. The central channel (6) is devoted to loading hematopoietic cells such as hematopoietic stem and progenitor cells (HSPCs), AML cell lines (MOLM-14 or NOMO-1), or AML patient blasts. The photography on the left shows the BMoC perfused with colorants to illustrate the position of the distinct compartments. Microscopy images on the right show centrosome, actin filaments and nucleus stainings within the endosteal and endothelial compartments. Scale bars correspond to 50 μm. (**B**) Immunostainings of the centrosome (red, pointed at with white arrowheads), the nucleus (blue) and the actin cytoskeleton (green) of cord blood (CB) HSPCs and AML cell lines, MOLM-14 and NOMO-1, interacting with osteoblasts (hFOBs) in the endosteal compartment. Scale bars correspond to 10 μm. (**C**) Quantification of the polarization index of CB HSPCs (red dots), MOLM-14 and NOMO-1 (blue dots) in contact with osteoblasts. Experiments were repeated three to five times and data were pooled ($n_{\text{CB HSPCs}} = 240$, $n_{\text{MOLM-14}} = 81$; $n_{\text{NOMO-1}} = 153$). Each spot corresponds to the measurement of the polarity of a single cell. The various shapes of the spots correspond to distinct experiments. Black bars show the median value and the standard deviation of the distribution. Differences between populations were evaluated using Kruskal–Wallis ANOVA test with $P$ values of 0.0002 and <0.0001 (***,****). (**D**) Same as (**B**) with CB HSPCs, MOLM-14 and NOMO-1 interacting with endothelial cells (HUVECs) in the BMoC endothelial compartment. Scale bars correspond to 10 μm. (**E**) Same as (**C**) with CB HSPCs, MOLM-14 and NOMO-1 interacting with endothelial cells (HUVECs) in the BMoC endothelial compartment. Experiments were repeated three times and data were pooled ($n_{\text{CB HSPCs}} = 106$, $n_{\text{MOLM-14}} = 167$; $n_{\text{NOMO-1}} = 215$). Differences between populations were evaluated using Kruskal–Wallis ANOVA test with $P$ values < 0.0001 and 0.0008 (****,***).

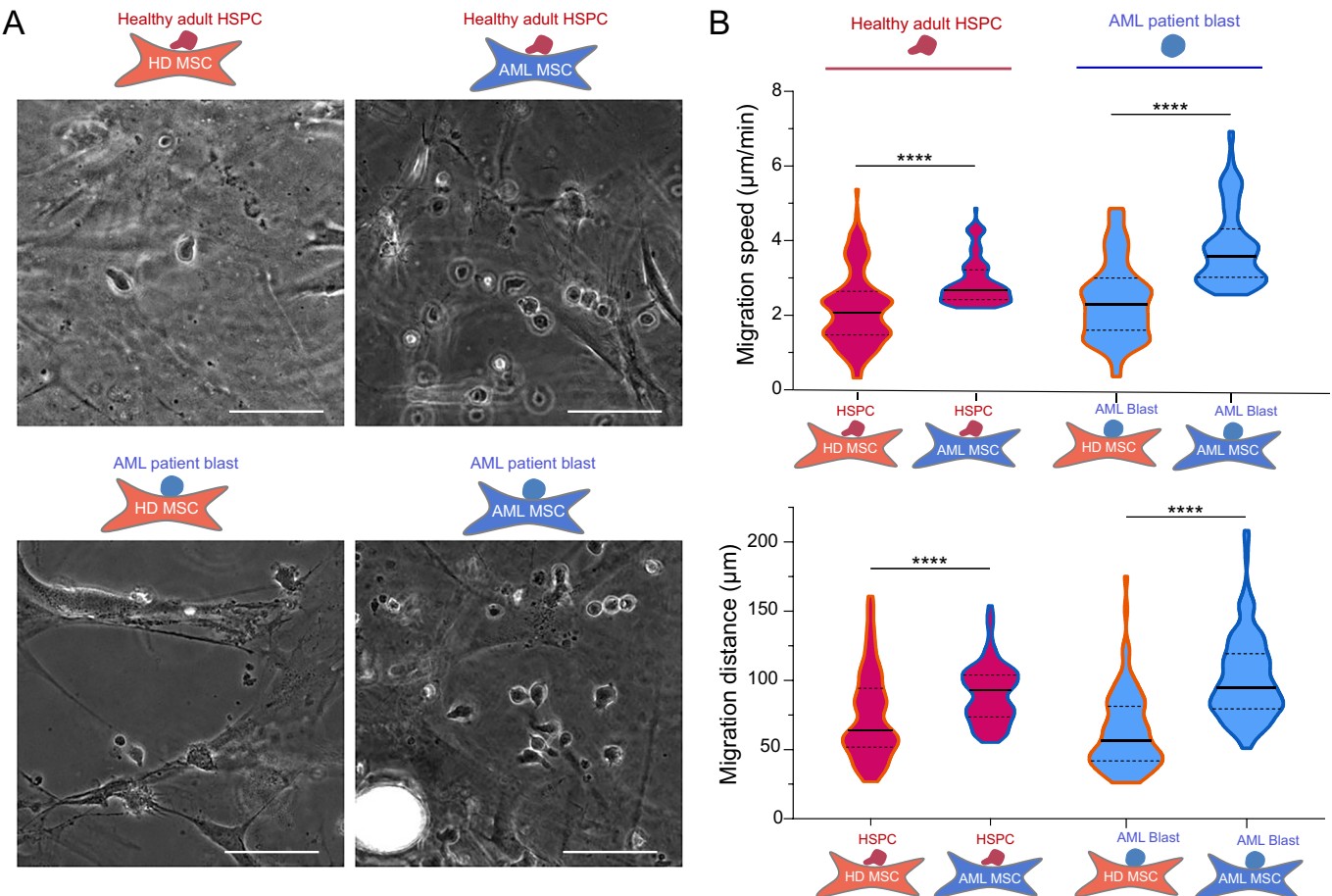

**Figure 5. Adult HSPCs and AML patient blasts migration in a humanized leukemic stromal compartment of a BMoC.**

(A) Instead of osteoblast, mesenchymal stem cells (MSCs) were collected from the bone marrow of a either healthy donor (orange in the schematics, left) or AML patient #5 and #6 (blue in the schematics, right) and loaded in the endosteal compartment of the BMoC (channel 3). Hematopoietic cells were collected from healthy adult (red in the schematics, top row) or AML patients #7 (blue in the schematics, bottom row). Images in transmitted light were extracted from time-lapse movies of healthy HSPC or AML patient blasts within healthy or leukemic compartments. Scale bars correspond to 50 μm. (B) Quantification of the mean speed and traveled distance during 20 min of the migration of healthy adult HSPCs (red) and AML blasts from patient #7 (blue) in a healthy (orange) or leukemic (blue) MSCs compartment. Tracks were analyzed at various positions within the stromal compartment ($n$ adult HSPC/ HD MSC = 88, $n$ adult HSPC/ AML MSC = 95, $n$ AML blast / HD MSC = 140, $n$ AML blast / AML MSC = 116). In the violin plots, black bars represent the median and dashed lines the 95% confidence interval. Differences between populations were evaluated using a Mann–Whitney test with $P$ values < 0.0001 (****).

CXCR4 pathway in the polarization and orientation of the division of cord blood-derived HSPCs (Bessy et al, 2021; Candelas et al, 2024). This pathway being involved in leukemic transformation (Ratajczak et al, 2006), it is thus likely to be involved in the reduced contact and defective polarization of leukemic blasts that we observed, as well as the numerous other adhesion pathways engaged in stem cell niche retention (Grenier et al, 2021).

The causal relationship between polarity defects and AML remains indeterminate, necessitating an exploration of the inter-relation between anchorage/polarity and proliferation/differentiation processes. Maintaining the balance between hematopoietic cell expansion and differentiation, a balance that is disturbed in AMLs, likely intricately hinges on cell polarization in interphase and asymmetric cell divisions, as it does in other stem cells (Neumüller and Knoblich, 2009; Sunchu and Cabernard, 2020). Hence, further examination is required, along with exploration of polarity together with the identity of progeny cells, through the defective segregation

of signals during leukemic cell division (Althoff et al, 2020; Mizukawa et al, 2017; Zimdahl et al, 2014; Candelas et al, 2024). In addition, cell polarization might be directly coupled to the control of cell cycle progression. Indeed, in multiple systems, the AMPK-LKB1 pathway connects cell polarity to the control of cell quiescence, by associating metabolic needs to the organization of cytoskeleton networks (Jansen et al, 2009; Kadekar et al, 2018). It might thus directly link HSC polarization to the balance between quiescence and proliferation (Florian and Geiger, 2010; Florian et al, 2012).

Noteworthy, the devised complementary engineered artificial niches present a methodological approach to quantify the functional behavior of blasts derived from patients, assessing the polarity across different stages of AML, with potential implications for future prognostic evaluations, as well as identification of the exact contribution of the loss of blast polarity to the progression of leukemia. They also constitute a convenient platform to parallelize

genomic or chemical screens combined with automated image analysis (Sockell et al, 2023) in order to reveal the molecular mechanism underlying the defective polarization of leukemic cells and identify chemical strategies to revert it.

## Methods

### Reagents and tools table

| Reagent/resource | Reference or source | Identifier or catalog number |
|---|---|---|
| **Experimental models** | | |
| Human umbilical cord blood cells | Cord Blood Bank of Public Assistance–Hospital of Paris (AP-HP), in coordination with the Biological Resources Centre authorized by the French Cord Blood Network. | N/A |
| Leftover hip bone fragments from healthy adult donors following hip replacement surgery | Lar-boisière Fernand-Widal AP-HP | N/A |
| AML patient blast samples | AML clinical registry with IRB approval no. IDRCB 2021-A00940-41 | N/A |
| MOLM-14 AML cell line | ATCC | #CRL-3006 |
| NOMO-1 AML cell line | DSMZ | #ACC-542 |
| HUVECs (Human Umbilical Vein Endothelial Cells) | Lonza | #CC-2517 |
| hFOB 1.19 osteoblast cell line | ATCC | #CRL-11372 |
| **Recombinant DNA** | | |
| **Antibodies** | | |
| Anti-human CD34 (Microbead Kit) | Miltenyi Biotec MACS | #130-046-702 |
| Polyclonal rabbit anti-pericentrin | Abcam | #ab4448 |
| Monoclonal mouse anti-pericentrin | Abcam | #ab47654 |
| Cross-absorbed secondary antibodies Alexa Fluor™ 568nm-conjugated goat anti-rabbit | Life Technologies | #A11004 |
| Cross-absorbed secondary antibodies Alexa Fluor™ 647nm-conjugated goat anti-mouse | Life Technologies | #A21240 |
| DAPI | Sigma-Aldrich | #D9542 |
| hCD34, APC, 581 | BC iotest | #IM2472 |
| hCD38, PeCy7, HB7 | BD Biosciences | #335825 |
| hCD45, V500, Hi30 | BD Biosciences | #560777 |
| hCD117, BV421, 104D2 | BD Biosciences | #563856 |
| hCD7, FITC, M-T701 | BD Biosciences | #332773 |
| hCD56, APC-A700, N901 NKH-1 | BC iotest | #B93446 |
| hCD13, PE, L138 | BD Biosciences | #332773 |

| Reagent/resource | Reference or source | Identifier or catalog number |
|---|---|---|
| hCD33, PerCPCy5.5, P67.6 | BD Biosciences | #333146 |
| HLA-DR, SNV786, C78087 | BC iotest | #Immu-357 |
| hCD19, APC-A750, J3-119 | BC iotest | #A94681 |
| hCD90, FITC, 5E10 | BD Pharmingen | #555595 |
| hCD97, PE, VIM3b | BD Pharmingen | #555774 |
| hCLL1, PE, 7D3 | BD Pharmingen | #563422 |
| hTIM3, PE, 50C1 | BD Pharmingen | #562566 |
| hCD45RA, APCH7, APCH7 | BD Biosciences | #560674 |
| hCD123, PE-Cy5.5, SSDCLY107D2 | BC iotest | #B20022 |
| hCD4, PE, 13B82 | BC iotest | #A07751 |
| hCD14, APC-A750, RM052 | BC iotest | #A86052 |
| hCD64, BV650, 10.1 | BD Biosciences | #740580 |
| hCD16, BV785, 3G8 | BD Biosciences | #563690 |
| **Oligonucleotides and other sequence-based reagents** | | |
| 94 myeloid gene Next Generation Sequencing (NGS) panel | Agilent SureSelect, Illumina | N/A |
| **Chemicals, enzymes, and other reagents** | | |
| EBM-2 | Lonza | #CC-3162 |
| EGM™ SingleQuots | Lonza | #CC-4176 |
| Ham's F12 DMEM (2,5 mM L-Glutamine/GlutaMAX™) | Gibco | #11514436 |
| Ficoll-Paque™ PLUS | Cytiva | #17144002 |
| Red blood lysis buffer | Sigma | #R7757 |
| RPMI 1640 medium | Thermo Fisher Scientific | #11875093 |
| Fetal bovine serum (FBS) | Sigma-Aldrich | #F2442 |
| Penicillin–streptomycin | Gibco, Life Technologies | #15070-063 |
| α-MEM medium | Thermo Fisher Scientific | #12571063 |
| Human Platelet Lysate (hPL) | StemCell Technologies | #100-610 |
| PDMS (polydimethylsiloxane) SYLGARD184™ - Kit | Dow Corning | #101697 |
| (3-Trimethoxysil) propyl-methacrylate | Sigma | #M6514 |
| Acetic acid | Sigma-Aldrich | #818755 |
| Acrylamide/bis-acrylamide 40% 37.5 | Euromedex | #EU0062-B |
| 2-hydroxy)2-methylpropiophenone (MPP) | Sigma-Aldrich | #405655 |
| Ammonium persulfate (APS) | Thermo Scientific | #17874 |
| tetra-methyl-ethylene-diamine (TEMED) | Sigma | #T9281 |
| Fibronectin | Sigma-Aldrich | #F1141 |
| Rat tail collagen type I | Ibidi | #50201 |
| Fibrinogen | Sigma-Aldrich | #F3879 |
| Thrombin | Sigma-Aldrich | #T4648 |
| Paraformaldehyde (16%) | Delta Microscopies | #D15710 |

| Reagent/resource | Reference or source | Identifier or catalog number |
|---|---|---|
| Triton X-100 | Sigma-Aldrich | #28103 |
| Bovine Serum Albumin (BSA) | Sigma-Aldrich | #A7030 |
| Tween-20 | Sigma-Aldrich | #P6585 |
| Mowiol | Sigma-Aldrich | #81381 |
| **Software** | | |
| Flow cytometry analysis software | FlowJo | #FlowJo v10 |
| Flow cytometry analysis software | Kaluza software | # v2.3, Beckman Coulter |
| Microfabrication design software | AutoCAD | #AutoCAD 2023 |
| Fiji + TrackMate7 plugin | Image J | #Image J 1.54f (Java 1.8.0_322) |
| GraphPad Prism 9 | GraphPad | Version 9 |
| **Other** | | |
| Nikon Biostation IM-Q Cell-S2 microscope | Nikon | Biostation interface 2.1. |
| Nikon Ti-eclipse microscope equipped with a spinning disc (Yokogawa-CSU-X1) with an electron-multiplying charge-coupled device camera (CCD camera Photometrics-Evolve512) | Nikon – Yokogawa-CSU-X1 | MetaMorph interface |
| Flow cytometer FACSDiva | BD Biosciences | #BD FACSDiva |
| Microfabricated quartz photomask | Manufactured by the lab using PRIMO Alvéole tech. | N/A |
| Bone Marrow on Chip (BMoC) | Outsourced private company: BeOnChip | N/A |
| UV LED exposure-masking system | Kloé | UV-KUB 2 |
| PE-50 plasma machine | Plasma ETCH Inc | |

## Hematopoietic cell purification

### Collection and processing of cord blood and bone fragments

Human umbilical cord blood cells were obtained from a compliant institution with ethical review board approval. Cord blood bags, provided by the Cord Blood Bank of Public Assistance–Hospital of Paris (AP-HP), were used in coordination with the Biological Resources Centre authorized by the French Cord Blood Network. Leftover hip bone fragments from adult donors to obtain adult HSPCs were collected following hip replacement surgery at Lariboisière Fernand-Widal AP-HP. Mononuclear cells were isolated using a density gradient, separating them from plasma, granulocytes, and erythrocytes using a lymphocyte separation medium (euro Bio, AbyCys). CD34$^+$ cells from cord blood or adult donors were sorted using microbeads conjugated to monoclonal mouse anti-human CD34 antibodies (Miltenyi Biotec MACS) through positive magnetic bead selection on LS columns.

### AML patient blast samples

**AML patient cell collection:** Blasts from AML patients were obtained from consenting individuals registered in an ongoing clinical registry at Saint Louis Hospital (THEMA, IRB approval no. IDRCB 2021-A00940-41). Cytogenetic analyses, karyotyping, and fluorescence in situ hybridization studies were conducted, as part of Saint Louis hospital standard of care clinical practice, along with genetic profiling using a Next Generation Sequencing (NGS) panel targeting 94 myeloid genes (ACD, ALDH2, ARID2, ASXL1, ASXL2, BCOR, BCORL1, BRAF, BRCA1, BRCA2, BRCC3, CALR, CBL, CCND1, CCND2, CDKN2A, CDKN2B, CEBPA, CHEK2, CREBBP, CSF3R, CSNK1A1, CTCF, CUX1, DDX41, DNM2, DNMT3A, EIF6, EP300, ERCC6L2, ETNK1, ETV6, EZH2, FLT3, GATA2, HRAS, IDH1, IDH2, IKZF1, IKZF5, IRF1, JAK2, JAK3, KDM5A, KDM6A, KIT, KMT2A/MLL, KMT2D, KRAS, LUC7L2, MBD4, MECOM, MGA, MPL, MPO, MYC, NF1, NPM1, NRAS, PDS5B, PHF6, PPM1D, PRPF8, PTEN, PTPN11, RAD21, RIT1, RUNX1, SAMD9, SAMD9L, SBDS, SETBP1, SETD1B, SF1, SF3B1, SH2B3, SMC1A, SMC3, SP1, SRP72, SRSF2, STAG2, TERC, TERT, TET2, TP53, TRIB1, U2AF1, U2AF2, UBA1, UBE2A, WT1, ZNF687, ZRSR2) to determine mutation status. Mononuclear cells were extracted from bone marrow samples using Ficoll-Paque™ PLUS (GE Healthcare #17-1440-02), followed by red blood cell lysis using a red blood lysis buffer (Sigma #R7757).

**Flow cytometry:** Following washing in PBS and resuspension in PBS containing 2 mM EDTA, hematopoietic cells from AML patients were characterized using fresh BM samples, employing a three-tube panel to identify the Leukemia Aberrant Immunophenotype (LAIP)/Different-from-Normal (Dfn) patterns, CD34$^+$CD38$^-$ leukemic stem cells (LSCs), and monocytic blast differentiation as described by Plesa et al (Plesa et al, 2024). Each tube included a core set of markers: CD34, CD38, CD45, and CD117. These were supplemented with additional markers tailored to specific analyses. For LAIP/Dfn characterization: CD7, CD56, CD13, CD33, HLA-DR, and CD19. For CD34 + CD38$^-$ LSCs: CD90 (Thy-1), a mix of CD97/CLL1/TIM3, CD45RA, and CD123. And for monocytic blast differentiation: CD4, CD14, CD64, and CD16. A total of one million cells were acquired per tube and analyzed using a DxFlex flow cytometer (Beckman Coulter). Data analysis was conducted using Kaluza software (v2.3, Beckman Coulter), with gating strategies for LAIP/Dfn and CD34$^+$CD38$^-$ LSCs presented in Fig. EV1 (Table 3). The complete list of antibodies utilized for FACS analysis, along with their references, is provided in the reagents and tools table.

### Hematopoietic cell culture conditions

The culture of hematopoietic cells, including cord blood/adult HSPC and AML patient blasts, involved maintenance in RPMI supplemented with fetal bovine serum (FBS), 1% of penicillin–streptomycin (5000 U/ml Penicillin, 5000 μg/ml Streptomycin (Gibco, Life Technologies #15070-063), and specific cytokines.

## Healthy donor and AML patient MSCs

Healthy donor MSCs were extracted from hip bone fragments of adult donors, while AML MSCs were cultured from bone marrow samples of AML patients. Before extracting mononuclear cells, a substrate of the bone marrow sample was cultured enabling MSCs

growth in α-MEM ((Stable Cell Minimum Essential Medium Eagle-α modification, Sigma #M6199) supplemented with human Platelet Lysate (hPL, StemCell#06960) and 1% penicillin–streptomycin.

## Cell lines

HUVECs (Lonza #00191027) and hFOB 1.19 (ATCC CRL-11372TM) were cultured in EBM-2 (Lonza #CC-3162) supplemented with EGM™ SingleQuots (Lonza #CC-4176), and in Ham's F12 DMEM (Dulbecco's Modified Eagle's Medium, 2.5 mM L-Glutamine/GlutaMAX™) supplemented with 10% FBS and 1% penicillin–streptomycin, respectively. MOLM-14 and NOMO-1 AML cell lines were cultured in RPMI (Roswell Park Memorial Institute 1640 medium) supplemented with 10% of FBS and 1% of penicillin–streptomycin.

Mycoplasma testing was conducted regularly, and cultures were maintained in a controlled environment.

## Microwell microfabrication and assembly

### Polyacrylamide microwells microfabrication

Microwell specifications, including shape, size, and layout, are designed using CleWin software to be subsequently transferred to a quartz photomask coated with a chromium layer (Toppan Photomask). They are fabricated on a silicon wafer. A negative mold of the silicon wafer is made in PDMS (polydimethylsiloxane SYLGARD184™—Dow Corning Kit). From a second positive PDMS stamp of microwells, stamps of $0.5 \times 0.5$ cm are cut.

Glass coverslips of $20 \times 20$ μm are coated with silane (2% (3-Trimethoxysil) propyl-methacrylate (Sigma #M6514)), and 1% acetic acid (Sigma-Aldrich #818755) in ethanol.

After a plasma treatment, the PDMS stamp is placed on a silanized coverslip pretreated with 2% (3-Trimethoxysil) propyl-methacrylate (Sigma #M6514)), and 1% acetic acid (Sigma-Aldrich #818755) in ethanol. A mixture containing 20% of a solution of polyacrylamide (acrylamide/bis-acrylamide 40% 37.5 (Euromedex #EU0062-B)), 1% of a photo-initiator (2-hydroxy)2-methylpropiophenone (MPP—Sigma-Aldrich #405655), 0.5% ammonium persulfate (APS, Thermo Scientific #17874), 0,5% tetra-methyl-ethylene-diamine (TEMED, Sigma #T9281) in MilliQ water is introduced between the PDMS and the coverslip by capillary forces. The mounted system is irradiated at 23 mJ/cm² for 5 min to cure the polyacrylamide gel and allow the stamp removal. For sterilization, microwells are irradiated in 70% ethanol, then rinsed in PBS overnight to remove any residual contaminants.

### Microwell assembly

Microwells were coated with fibronectin (10 μg/ml, Sigma #F1141). After PBS rinse, stromal cells [hFOB–HUVEC –HD MSC–AML patient MSC] were added in their adequate media. After incubation, hematopoietic cells [CB/adult CD34⁺ HSPC–MOLM-14–NOMO-1–AML patient blasts] were loaded and incubated in their regular media.

## Bone-marrow-on-chip assembly

### Microfabrication and outsourcing

The chip design as reported in previous lab works (Bessy et al, 2021; Souquet et al, 2021), inspired by the Noo Li Jeon laboratory, faced limitations in homemade manufacturing, leading to outsourcing to Biomimetic Environment On Chip company. The chips underwent UV treatment for sterilization before use.

### Hydrogel composition and cell loading

A solution of rat tail collagen type I (Ibidi #50201) at 1.6 mg/ml, fibrinogen at 2 mg/ml (Sigma #F3879), and thrombin at 1 U/ml (Sigma # T6884) compose the hydrogel encapsulating stromal cells. In total, $2.10^5$ HUVECs or hFOBs or HD MSCs or AML MSCs are loaded in the gel within the dedicated channels.

The BMoC is then incubated for gelation at 37 °C for ~45 min. Regular media of the stromal cells are injected into the medium inlets.

When the stromal cell compartments are functional (~48 h) cord blood and adult CD34⁺ HSPCs, AML cell lines, or AML patient blasts can be loaded in the central channel of the chip. All medium inlets are subsequently filled with the regular medium of the hematopoietic cells.

## Immunostaining

Cells within microwells or chips were fixed (4% paraformaldehyde (16% PFA, Delta Microscopies #D15710)), permeabilized (0.1% Triton X-100 (Sigma-Aldrich)), and blocked (3% BSA (Bovine Serum Albumin, Sigma-Aldrich #A7030) and 0.1% Tween (Sigma-Aldrich)). Primary antibodies included polyclonal rabbit anti-pericentrin (Abcam #ab4448) at 1/600, and monoclonal mouse anti-pericentrin (Abcam #ab47654) at 1/200. Cross-absorbed secondary antibodies Alexa Fluor™ 568 nm- and 647 nm-conjugated goat anti-rabbit and anti-mouse (Life Technologies #A11004 and #A21240) were used at 1/600. F-actin was labeled with Alexa Fluor 488-conjugated phalloidin (Sigma-Aldrich #A12379). Nucleus localization is stained using DAPI (Sigma-Aldrich #D9542). The coverslips containing microwells are mounted in mowiol after the removal of polyacrylamide microwell walls (Sigma-Aldrich #81381). The BMoCs are mounted in Mowiol added within their inlets and outlets.

## Microscopy

### Live and immunofluorescence imaging

Time-lapse imaging for BMoC required the use of the Nikon Biostation IM-Q Cell-S2 microscope (Institute of Research of Saint Louis platform). After loading with hematopoietic cells, the devices are placed in an enclosed chamber at 37 °C with controlled humidity and normoxic conditions to be imaged every 2 min. Image processing is performed with 20 × 0.80-NA air objective and binnings of $2 \times 2$, using the Biostation interface 2.1.

Images of fixed samples are captured using a Nikon Ti-eclipse microscope equipped with a spinning disc (Yokogawa-CSU-X1) and a 100×1.40-NA oil objective, together with an electron-multiplying charge-coupled device camera (CCD camera Photometrics-Evolve512). The acquisition software used is MetaMorph. Z-stacks of 0.5μm thickness were acquired using a $2 \times 2$ binning.

### Polarization index measurement

Centrosome position was measured using a MACRO via the Fiji interface. First, the centrosome position is manually marked. Second, the membrane zone of the cell of interest where it interacts

with the stromal cell is marked to record the distance to the centrosome (distance d). Finally, a third point representing the localization of the tip of the cell is selected as the point in the cell furthest away from the previous one. The polarization index is calculated as the ratio of the distance from the centrosome to the contact zone, to the total length of the cell (d/D).

### Cell tracking

Cell tracking is performed using TrackMate7 an open-source platform, distributed in Fiji, for single-particle tracking, data visualization and editing results (Ershov et al, 2022). Relying on three filters on tracks, track duration, total displacement, and mean speed are set to compare tracks displaying the same time.

## Statistics

All of the statistical analyses were carried out using GraphPad Prism 9 software. A Shapiro–Wilk test is used to assess the normality of the data distribution and to guide the selection of appropriate statistical tests. Differences between populations are assessed using non-parametric methods such as Kruskal–Wallis ANOVA and Mann–Whitney $t$ test for nonnormally distributed samples, or parametric tests such as unpaired $t$ test and one-way ANOVA for normally distributed samples. All data are presented as superplot groups with different biological replicates, each represented by a distinct spot shape.

## Data availability

Source data were deposited on Bioimage Archive. The accession number is S-BIAD1681.

The source data of this paper are collected in the following database record: biostudies:S-SCDT-10_1038-S44319-025-00466-w.

## Peer review information

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

## Acknowledgements

We express our gratitude to A Puissant and R Itzykson (Saint Louis Hospital, U944, Université Paris Cité) for their valuable scientific insights. Our appreciation extends to Niclas Setterblad from the Saint Louis Research Institute Core Facility at Saint Louis Hospital. The work directed by Manuel Théry was funded by the French National Research Agency (AAPG2022-PRC-SHARP and ANR-23-CHBS-0013), the Bettencourt-Schueller Foundation, the Schlumberger Foundation for Education and Research, the National Center for Precision Medicine in Leukemia (THEMA) at the Saint Louis Research Institute and Bristol Myers Squibb. The work led by Lina Benajiba was funded by the Institut National du Cancer (PRTK 2022-192), the Ligue Nationale Contre le Cancer, the ATIP-Avenir program of the Bettencourt-Schueller Foundation, the "InsiTu" Integrated Cancer Research Site and the Institut de la Leucémie Paris Saint Louis.

## Author contributions

**Khansa Saadallah**: Conceptualization; Data curation; Formal analysis; Validation; Investigation; Methodology; Writing—original draft; Writing—review and editing. **Benoît Vianay**: Investigation; Methodology. **Louise Bonnemay**: Investigation; Methodology. **Hélène Pasquer**: Formal analysis; Methodology. **Lois Kelly**: Resources. **Stéphanie Mathis**: Formal analysis; Methodology. **Cécile Culeux**: Resources. **Raphael Marie**: Resources. **Paul Arthur Meslin**: Formal analysis; Methodology. **Sofiane Fodil**: Resources. **Paul Chaintreuil**: Resources. **Emeline Kerreneur**: Resources. **Arnaud Jacquel**: Resources. **Emmanuel Raffoux**: Resources. **Rémy Nizard**: Resources. **Camille Lobry**: Formal analysis; Methodology. **Laurent Blanchoin**: Supervision; Funding acquisition; Validation; Project administration. **Lina Benajiba**: Resources; Funding acquisition; Validation; Writing—review and editing. **Manuel Théry**: Conceptualization; Supervision; Funding acquisition; Validation; Writing—original draft; Writing—review and editing.

Source data underlying figure panels in this paper may have individual authorship assigned. Where available, figure panel/source data authorship is listed in the following database record: biostudies:S-SCDT-10_1038-S44319-025-00466-w.

## Disclosure and competing interests statement

The authors declare no competing interests.

# Expanded View Figures

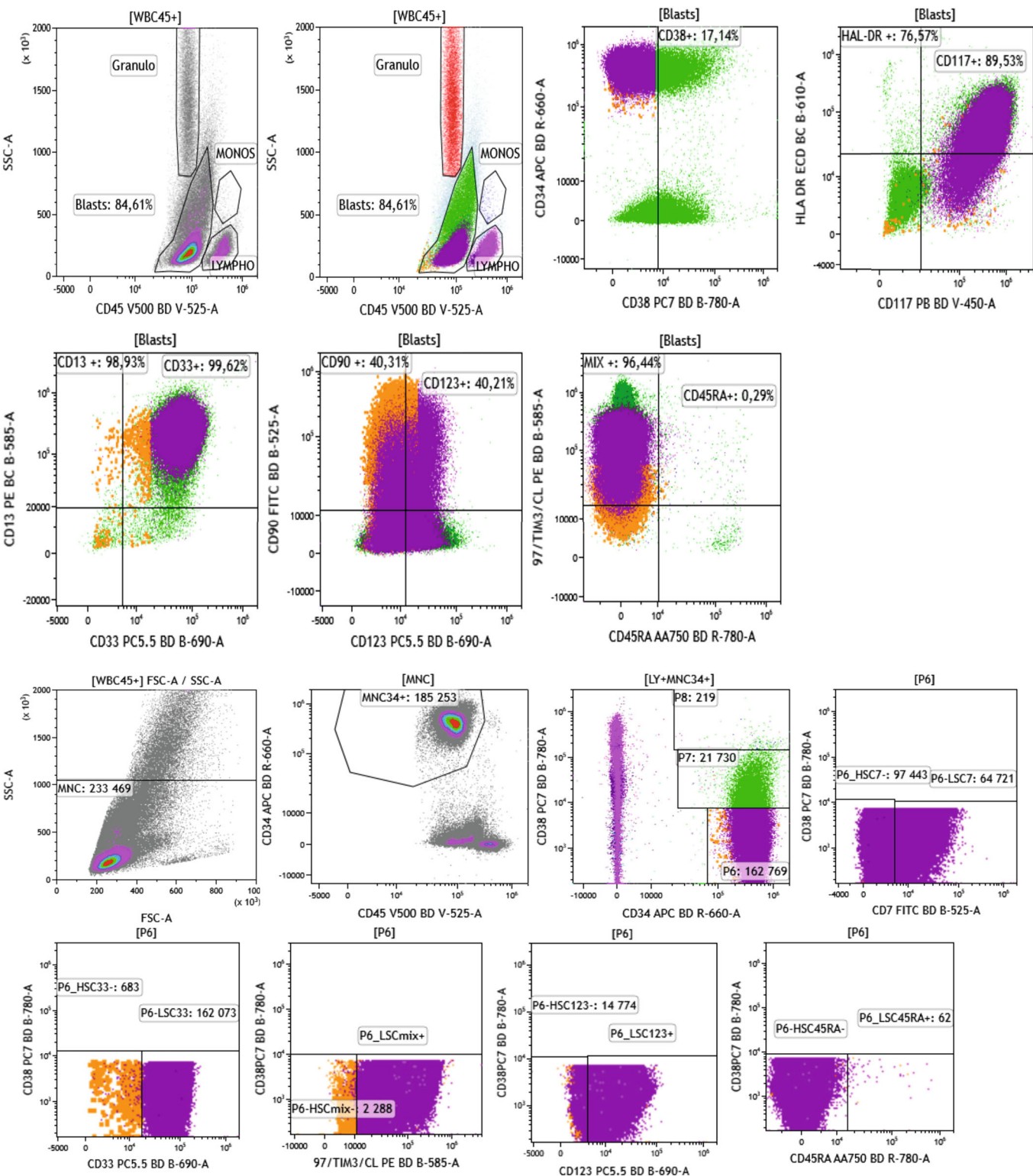

**Figure EV1. AML patient n° 10 hematopoietic cell flow cytometry gating strategy (blasts and LSCs).**

Flow cytometry analysis of an AML patient with a high LSC fraction.

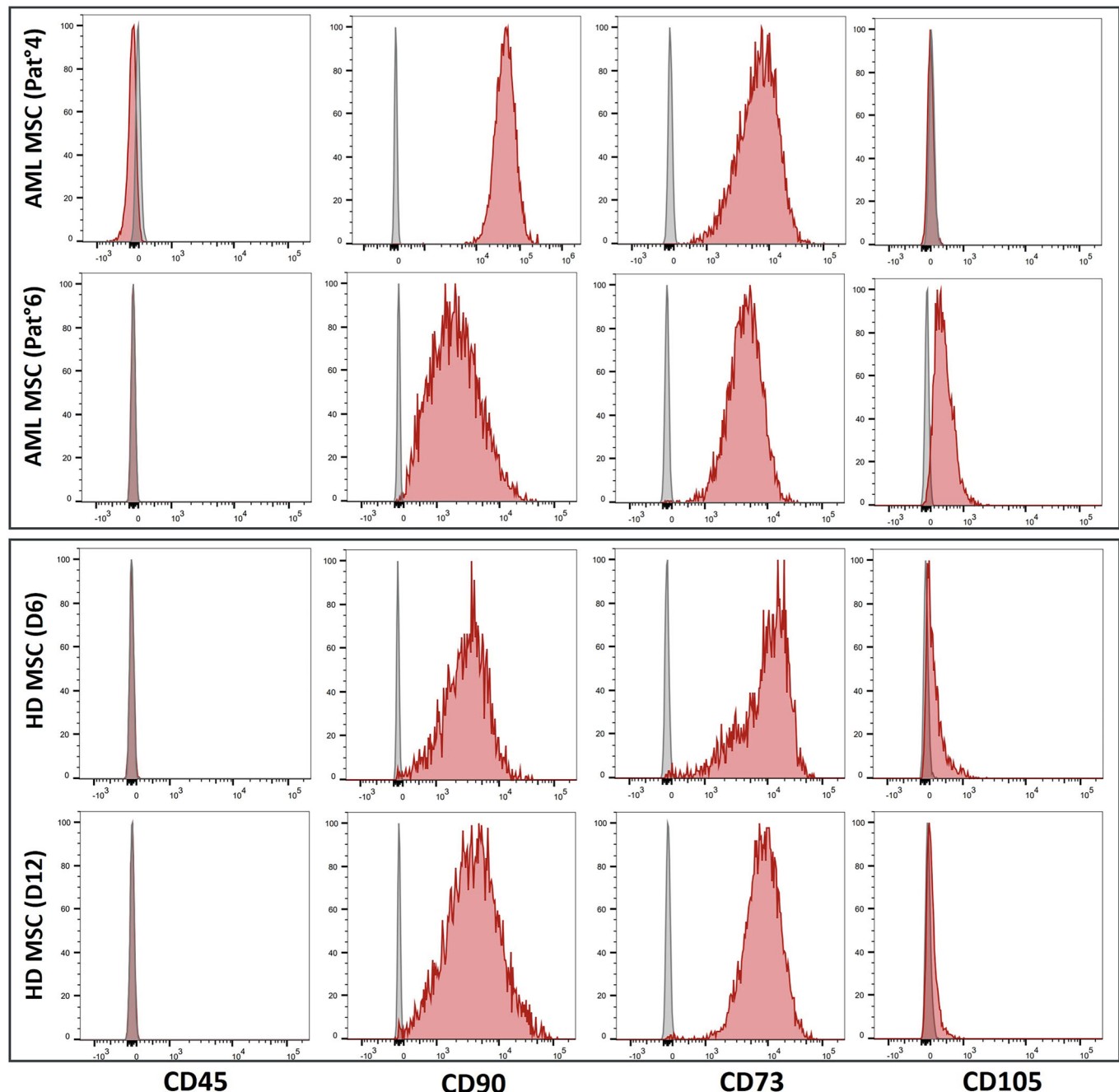

**Figure EV2.  AML patient immunophenotypic profiles of mesenchymal stromal cells (MSCs).**

Flow cytometry strategy to characterize from the AML patient MSCs considered as CD45, CD90, CD73, CD105.

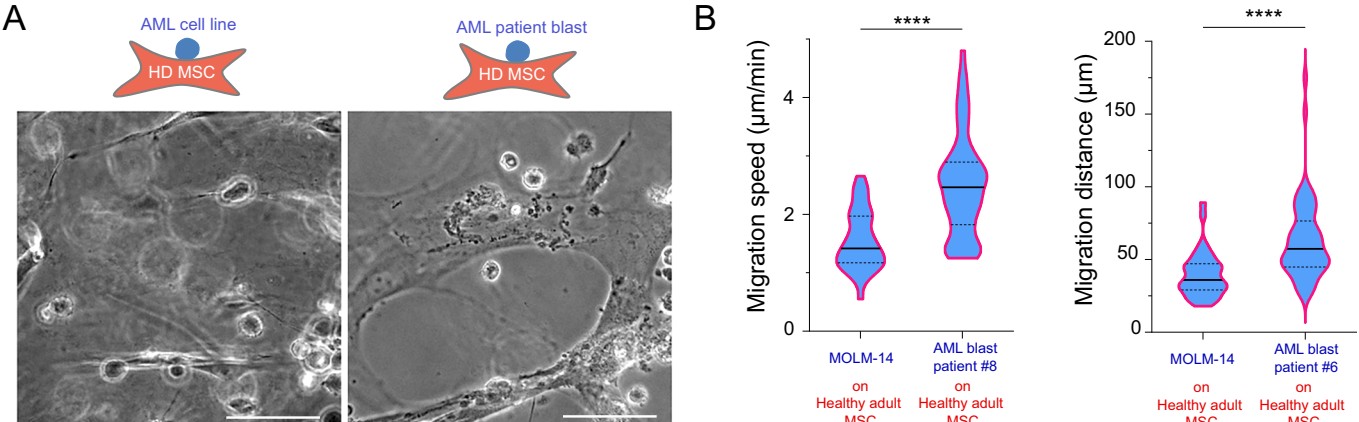

**Figure EV3. Leukemic cells originating from AML cell lines lack the motility observed in AML patient blasts within a healthy mesenchymal stromal cell compartment in the BMoC model.**

(A) AML cell line (MOLM-14) and AML patient blasts (blue in the schematics) were loaded in the BMOC and thus interacted with MSC from healthy donors (orange in the schematics). Images in transmitted light were extracted from time-lapse movies of AML cell lines and patient blast within healthy compartments. Scale bars correspond to 50 μm. (B) Quantification of the mean speed and traveled distance during 20 min of the migration of leukemic cells (MOLM-14) or AML patient blast in a healthy stromal compartment (HD MSC). Tracks were analyzed at various positions within the stromal compartment ($n_{MOLM-14/\ HD\ MSC}$ = 41, $n_{AML\ blast/\ HD\ MSC}$ = 77). In the violin plots, black bars represent the median and dashed lines the 95% confidence interval. Differences between populations were evaluated using a Mann–Whitney test with $P$ values < 0.0001 (****).

