## [Peer Review File · EMBO Reports]

AML patient blasts exhibit polarization defects upon interaction with bone marrow stromal cells.

Khansa Saadallah, Benoit Vianay, Louise Bonnemay, Helene Pasquer, Lois Kelly, Stephanie Mathis, Cecile Culeux, Raphael Marie, Paul Meslin, Sofiane Fodil, Paul Chaintreuil, Emeline Kerreneur, Arnaud Jacquel, Emmanuel Raffoux, Remy Nizard, Camille Lobry, Laurent BLANCHOIN, Lina Benajiba, and Manuel Thery

Corresponding author(s): Manuel Thery (manuel.thery@cea.fr)

Review Timeline:

Submission Date:	5th Jul 24
Editorial Decision:	16th Aug 24
Revision Received:	14th Feb 25
Editorial Decision:	20th Mar 25
Revision Received:	10th Apr 25
Accepted:	17th Apr 25

Editor: Achim Breiling

Transaction Report:

Dear Dr. They,

Thank you for the submission of your manuscript to EMBO reports. I have now received the reports from the three referees that were asked to evaluate your study, which can be found at the end of this email.

As you will see, the referees find the study interesting, but they have several comments, concerns, and suggestions that need to be addressed to allow publication of the study in EMBO reports. EMBO reports emphasizes striking novel functional over detailed mechanistic insight, thus we will not require addressing points regarding more mechanism or causalities experimentally. However, it will be necessary that during revision you address all points questioning the main conclusions of the study, and all technical concerns, or points regarding the experimental designs, model systems used, or data presentation and discussion.

Given the constructive referee comments, I would thus like to invite you to revise your manuscript with the understanding that the concerns of the referees must be addressed in the revised manuscript (as indicated above) or in a detailed point-by-point response.

Acceptance of your manuscript will depend on a positive outcome of a second round of review. It is EMBO reports policy to allow a single round of revision only and acceptance of the manuscript will therefore depend on the completeness of your responses included in the next, final version of the manuscript.

1) a .docx formatted version of the final manuscript text (including legends for main figures, EV figures and tables), but without the figures included. Figure legends should be compiled at the end of the manuscript text.

2) individual production quality figure files as .eps, .tif, .jpg (one file per figure), of main figures and EV figures. Please upload these as separate, individual files upon re-submission.

We would publish your manuscript in the Report format. For a Scientific Report we allow up to 5 main and EV figures and require that results and discussion sections are combined in a single chapter called "Results and Discussion". Please do this for your manuscript and also add section headings to this part. For more details, please refer to our guide to authors:

<http://www.embopress.org/page/journal/14693178/authorguide#researcharticleguide>

4) a complete author checklist, which you can download from our author guidelines (<https://www.embopress.org/page/journal/14693178/authorguide>). Please insert page numbers in the checklist to indicate where the requested information can be found in the manuscript. The completed author checklist will also be part of the RPF.

Please also follow our guidelines for the use of living organisms, and the respective reporting guidelines: <http://www.embopress.org/page/journal/14693178/authorguide#livingorganisms>

5) that primary datasets produced in this study (e.g. RNA-seq, ChIP-seq, structural and array data) are deposited in an appropriate public database. If no primary datasets have been deposited, please also state this in a dedicated section (e.g. 'No primary datasets have been generated and deposited'), see below.

The accession numbers and database should be listed in a formal "Data Availability" section (placed after Materials & Methods) that follows the model below. This is now mandatory (like the COI statement). Please note that the Data Availability Section is restricted to new primary data that are part of this study. This section is mandatory. As indicated above, if no primary datasets have been deposited, please state this in this section

Data availability

8) Regarding data quantification and statistics, please make sure that the number "n" for how many independent experiments were performed, their nature (biological versus technical replicates), the bars and error bars (e.g. SEM, SD) and the test used to calculate p-values is indicated in the respective figure legends (also for EV figures and all those in an Appendix). Please also check that all the p-values are explained in the legend, and that these fit to those shown in the figure. Please provide statistical testing where applicable. Please avoid the phrase 'independent experiment', but clearly state if these were biological or technical replicates. Please also indicate (e.g. with n.s.) if testing was performed, but the differences are not significant. In case n=2, please show the data as separate datapoints without error bars and statistics. See also: <http://www.embopress.org/page/journal/14693178/authorguide#statisticalanalysis>

9) Please add scale bars of similar style and thickness to microscopic images, using clearly visible black or white bars (depending on the background). Please place these in the lower right corner of the images themselves. Please do not write on or near the bars in the image but define the size in the respective figure legend.

10) Please also note our reference format:

11) We updated our journal's competing interests policy in January 2022 and request authors to consider both actual and perceived competing interests. Please review the policy <https://www.embopress.org/competing-interests> and update your

competing interests if necessary. Please name this section 'Disclosure and Competing Interests Statement' and put it after the Acknowledgements section.

12) We now use CRediT to specify the contributions of each author in the journal submission system. CRediT replaces the author contribution section. Please use the free text box to provide more detailed descriptions and do NOT provide your final manuscript text file with an author contributions section. See also our guide to authors:
<https://www.embopress.org/page/journal/14693178/authorguide#authorshipguidelines>

13) All Materials and Methods need to be described in the main text using our 'Structured Methods' format, which is required for all research articles. According to this format, the Materials and Methods section should include a Reagents and Tools Table (listing key reagents, experimental models, software, and relevant equipment and including their sources and relevant identifiers), uploaded as separate file, followed by a Methods and Protocols section in which we encourage the authors to describe their methods using a step-by-step protocol format with bullet points, to facilitate the adoption of the methodologies across labs. More information on how to adhere to this format as well as downloadable templates (.doc) for the Reagents and Tools Table can be found in our author guidelines (section 'Structured Methods'):

14) Please add up to 5 keywords and order the manuscript sections like this, using these names:
Title page - Abstract - Keywords - Introduction - Results & Discussion - Methods - Data availability section - Acknowledgements - Disclosure and Competing Interests Statement - References - Figure legends - Expanded View Figure legends

I look forward to seeing a revised form of your manuscript when it is ready.

Yours sincerely,

Referee #1:

In this manuscript, the authors aim to investigate the polarization of AML patient blasts and the impact of AML patient-derived stromal cells on the polarization of healthy HSPCs. Utilizing microwells and a bone-marrow-on-a-chip model, the authors identified polarization defects in AML patient blasts. An impressive piece of work, though there are several major issues with the manuscript.

Major issues:

1. The authors claim that their results demonstrate an association between leukemia progression and defective polarization of AML blasts. Indeed, their findings reveal polarization defects in AML blasts and increased motility induced by AML-derived stromal cells. However, the manuscript does not explore how these changes contribute to leukemia progression or identify the specific mechanisms involved in these defects. To support this claim, it would be beneficial to investigate whether these defects hinder HSPC expansion, self-renewal, or cause exhaustion, and whether these defects promote AML blast proliferation or enable AML blasts to outcompete healthy HSPCs.

2. Several conclusions in the manuscript are confusing. The authors demonstrate a loss or decrease in polarization of HSPCs and AML blasts, the absence of magnupodium in AML blasts, and faster migration of AML blasts. These observations suggest reduced adhesion and/or association between AML blasts and stromal cells. However, previous publications (PMID: 16998484, 16998483) reported increased expression of CD44 in LSCs, which facilitated homing, engraftment, and adhesion. It would be beneficial to further examine changes in adhesion molecules, cadherins, receptors, or integrins and conduct genetic studies to validate these findings.

Minor issues:

1. Page 4 paragraph 3. ".....MSCs derived from healthy donors or AML patients (Figure 3. A. c-f)....." Please check if a-f is the correct reference.
2. Page 4 paragraph 5. "We first tested the ability of HSPCs to polarize tin contact with either endosteal (Figure 4.B.a-c) or endothelial cells (Figure 4.C.a-c)....." It seems Figure 4B.b-c and Figure 4C.b-c are not referred to HSPC.
3. For the polarization index, it would be helpful to label the "d" and "D" in Figures to clarify how the measurement was taken. Additionally, specify whether the cross-sectional or front view was used, as the orientation of some cells is unclear.

Referee #2:

In this article Saadallah et al. use microphysiological devices to determine the polarization of primary human AML blasts and normal HSPCs when co-cultured with MSCs derived from normal and leukemia bone marrow biopsies. While this type of analysis is technically challenging and of general interest, the data presented is fairly descriptive and lacks mechanistic depth. In particular, the physiological significance of differences in polarization of normal and leukemic cells in this co-culture system is unclear.

Major Comments

1. Given that stromal cells are known to support both normal human HSPCs as well as leukemic cell survival and chemoresistance, the physiological relevance of differences in the way the leukemic cells polarize compared to HSPCs in these systems is unclear.
2. Does leukemic and normal cell polarization (or no polarization) towards or away from stromal cells as shown in figure 2 alter the fate of the daughter cells post cell division? Thus, would there be a shift in symmetric vs. asymmetric cell division?
3. Do magnupodia enrich for any known markers of cell polarity? Do the HSPCs and AML cells co-cultured with stromal cells display polarized expression of Cdc42 and/or Numb?
4. While the methods indicate that single HSPCs/AML cells were seeded along with single MSCs, the images shown in Figs. 1-3 seem to indicate that more than one hematopoietic cell was present in many cases. This is likely due to the technical challenges associated with single cell seeding. However, this does raise the question whether stromal cell contact with two or more hematopoietic cells alter polarization as compared to one-one cell contact?
5. It would be helpful to include normal differentiated cells as controls to determine if lack of polarization is specific to the leukemic state and not simply a readout of progenitor/mature cells behavior, especially since the LSC content in all AML samples used is fairly low. In addition, data should be included from AML samples enriched for CD34+ cells as is done for normal bone marrow.

Minor Comments:

1. The references have several discrepancies that need to be corrected.

Referee #3:

The manuscript "AML patient blasts exhibit polarization defect upon interaction with bone marrow stromal cells" by Saadallah K. et al, investigates the capacity of leukemic stem cell (LSC) to polarize upon contact with the niche cells.

The investigation of LSC polarity upon contact with niche cells is novel and interesting. However, the data included in the current version of the manuscript requires some important clarifications and/or some additional experiments are needed. The major points to be addressed are:

- 1) The role of polarity in AML has been associated with AML development and survival. Cell polarity is important for cell migration/motility but in the context of AML it is somehow not so critical as cell proliferation or AML survival/development. Can the authors investigate leukemic cell proliferation/survival alongside polarization?
- 2) Another important aspect is related with the overall study design, where the authors compare CD34+ HSPCs from healthy donor with unenriched mononuclear cells from AML patients (with very high blast counts). The leukemic blasts used in the study must be better characterized by flow cytometry. Ideally it would be more relevant to compare LSC vs HSPCs.
- 3) Along the same line, also the healthy and leukemic niche should be better characterized by providing some flow cytometry analysis of the niche cells used for the polarity assay.
- 4) It is likely that each patient has different frequency and composition of LSC and niche cells. How do the authors take this aspect into consideration?
- 5) The characteristics (number, age, sex etc) of the healthy donor cohort are not provided.
- 6) In figure 4 the same picture is used in panel Ad and Bb with different labels.
- 7) The last sentence of the abstract is an overstatement. There is no data in the manuscript supporting this claim because there is no analysis of leukemic progression nor of the causality between defective polarization of AML blasts and leukemic progression.

We sincerely thank the reviewers for their thoughtful evaluations of our manuscript and their constructive feedbacks. Their overall assessments were similar in a number of respects. They appreciated the novelty of the description of the loss of polarity in AML blasts and were excited by the use of artificial micro-niches, which were considered innovative and potentially applicable to many other studies of blast behavior. They also wondered if this loss of polarity played a role in the progression of AML. Although we felt that this fair comment would take us beyond the scope of the current study, we agreed that this was an important point and did our best to move in this direction. Since blasts obviously contribute to leukemia progression, the first step was to identify the molecular pathway involved in HSC polarization that is disrupted in blasts in order to modulate it in healthy HSCs and assess the specific impact of their defective polarity on leukemia progression. We performed a transcriptomic analysis of all blasts used in the study. We obtained a heat map showing that, in addition to the expected genes involved in proliferation and differentiation, some genes involved in cell migration, adhesion and polarity were also up- or down-regulated in the blasts. Unfortunately, the inter-patient variability did not allow the unambiguous identification of defective polarity pathways. A deeper and more focused study with more patients of the same age and at the same stage of disease progression seemed necessary. We regret not being able to better address this key point. We hope that the reviewers will consider that the discovery of the loss of the ability of AML blasts to polarize towards the stromal cells of their niche is sufficiently novel and significant to merit publication in EMBO Reports.

We have addressed all the other comments raised by the reviewers and incorporated additional data, including new supplementary materials and a novel experiment added to the previous figures. These experiments were designed to:

- (1) investigate the polarity of leukemic stem cells (LSC). We took advantage of a recent sample from a new patient with a high percentage LSCs. The analysis confirmed the defective polarization of LSCs.
- (2) provide a detailed characterization of the AML patient cells with advanced flow cytometry techniques.
- (3) investigate the potentially polarized distributions of cell determinant such as Numb and Cdc42.

Overall, we believe the additional data and revisions have contribute to strengthen the manuscript, and we hope the reviewers now consider it suitable for publication in EMBO Reports.

Our detailed responses to the specific comments are provided below:

Referee #1

In this manuscript, the authors aim to investigate the polarization of AML patient blasts and the impact of AML patient-derived stromal cells on the polarization of healthy HSPCs. Utilizing microwells and a bone-marrow-on-a-chip model, the authors identified polarization defects in AML patient blasts. An impressive piece of work, though there are several major issues with the manuscript.

Major issues:

1. The authors claim that their results demonstrate an association between leukemia progression and defective polarization of AML blasts. Indeed, their findings reveal polarization defects in AML blasts and increased motility induced by AML-derived stromal cells. However, the manuscript does not explore **how these changes contribute to leukemia progression** or **identify the specific mechanisms** involved in these defects. To support this claim, it would be beneficial to investigate whether these **defects hinder HSPC expansion, self-renewal, or cause exhaustion**, and **whether these defects promote AML blast proliferation or enable AML blasts to outcompete healthy HSPCs**.

We fully agree with this comment. The question of the molecular mechanism involved in HSC polarization and the impact of its impairment on blast proliferation/differentiation and leukemia progression was raised by the three reviewers. We invite the reviewer to read the joint response on the first page of this document.

To start investigating the molecular mechanism involved in HSC polarization, we performed a transcriptomic analysis of all the patient samples we used in the study. Here is the heatmap of the main adhesion and polarity genes displaying differential expression between healthy CD34+ cord blood HSPC (CB) and AML patient samples (blast and LSC). As expected, CB exhibited lower expression of proliferation-related genes (CCNE1, CCND1, CCNB1, CDKN2C, E2F1, MIK67 and PCNA) as compared to leukemic blasts and more particularly to LSCs. Genes related to cell adhesion, cytoskeleton and polarity genes displayed less obvious changes. Some variations such as tubulin and stathmin (microtubule stabilization), or Prominins and Numb (cortical polarity markers), may indicate some genuine differences but these would need more samples to be confirmed since the statistical analyses of the current data set did not validate their significance. Cdc42 displayed large inter-sample variations in both CB and blasts.

2. Several conclusions in the manuscript are confusing. The authors demonstrate a loss or decrease in polarization of HSPCs and AML blasts, the absence of magnupodia in AML blasts, and faster migration of AML blasts. These observations **suggest reduced adhesion and/or association between AML blasts and stromal cells**. However, previous publications (PMID: 16998484, 16998483) reported **increased expression of CD44 in LSCs**, which **facilitated homing, engraftment, and adhesion**. It would be beneficial to further examine changes in **adhesion molecules, cadherins, receptors, or integrins** and conduct **genetic studies to validate these findings**.

The faster migration and reduced polarization observed in AML blasts, coupled with the absence of magnupodia, indeed suggest altered adhesion compared to healthy HSPCs.

While previous studies, including those referenced by the reviewer, have shown elevated CD44 expression in LSCs that supports homing and engraftment, our findings provide a nuanced perspective that highlights differential behaviors within the leukemic hematopoietic cell hierarchy. As properly mentioned by the reviewer, elevated CD44 expression in LSCs facilitates homing and engraftment by binding to hyaluronic acid (HA) and other stromal components, promoting initial adhesion to the bone marrow niche (Jin et al., Nat Rev Cancer, 2006). The interaction between CD44 and HA facilitates also their maintenance in a quiescent state. However, through its binding to L-selectin on endothelial cells, CD44 also activates the small GTPases Rac and Rho, and thereby promotes their transition toward a migratory phenotype (Zöller 2015). So the exact role of CD44 in LSC is unclear. Furthermore, AML blasts represent a different population with respect to their capacity for adhesion and migration. Depending on the disease stage and underlying mutational profile, AML blasts may adopt mechanisms favoring cell migration and dissemination rather than adhesion (Gruszka et al. 2019). Loss of polarization and magnupodia in AML blasts may thus reflect a transition to a motile phenotype, consistent with reduced niche dependence.

To clarify the potential role of CD44, we performed a transcriptomic study of all blasts used in our study. They showed low levels of CD44 expression (see previous answer). This is consistent with their low adhesion, defective polarity, and faster migration. However, patient 10 showed a higher proportion of LSCs, increased CD44 expression compared to the AML blasts, and also defective cell polarization, suggesting that CD44 expression is not directly controlling cell polarization. We analyzed the transcriptomic profiles of all patients and could not identify a specific adhesion pathway that was highly defective in all non-polarized blasts and present in all polarized healthy HSCs. This suggests that cell polarity depends on a complex combination of adhesion and signaling pathways, the elucidation of which would require a much larger set of patient samples, which seemed beyond the scope of our study.

Minor issues:

1. Page 4 paragraph 3. ".....MSCs derived from healthy donors or AML patients (Figure 3. A. c-f)....." Please check if a-f is the correct reference.

We thank the reviewer for paying attention to this mistake. We corrected it.

2. Page 4 paragraph 5. "We first tested the ability of HSPCs to polarize tin contact with either endosteal (Figure 4.B.a-c) or endothelial cells (Figure 4.C.a-c)....." It seems Figure 4B.b-c and Figure 4C.b-c are not referred to HSPC.

We thank the reviewer for paying attention to this mistake. We corrected it.

3. For the polarization index, it would be helpful to label the "d" and "D" in Figures to clarify how the measurement was taken. Additionally, specify whether the cross-sectional or front view was used, as the orientation of some cells is unclear.

Actually, measurements are not done on a cross-section or projected view. They are true 3D distance. Please refer to the "Polarization Index Measurement" section in the Materials and Methods for a detailed description of the methodology used to measure the distance from the centrosome to the contact site relative to the total cell length. The macro implemented through the Fiji interface accounts for cell orientation by measuring distances along the axis, utilizing z-stack imaging with slices separated by 0.3 μm .

Referee #2

In this article Saadallah et al. use micro-physiological devices to determine the polarization of primary human AML blasts and normal HSPCs when co-cultured with MSCs derived from normal and leukemia bone marrow biopsies. While this type of analysis is technically challenging and of general interest, the data presented is fairly descriptive and **lacks mechanistic depth**. In particular, the **physiological significance of differences in polarization** of normal and leukemic cells in this co-culture system is unclear.

Major Comments

1. Given that stromal cells are known to support both normal human HSPCs as well as leukemic cell survival and chemoresistance, the **physiological relevance of differences in the way the leukemic cells polarize compared to HSPCs in these systems is unclear**.

This is a fair comment. As stated in the introduction, polarity defects have been associated to the progression of several solid tumors, contributing to both cell proliferation, tissue disorganization and cell migration. Our work is the first to show similar defects in blood cancer. However, we agree that our observations do not demonstrate that these changes play a causal role in the progression of leukemia. This concern was raised by the three reviewers. We invite the reviewer to read the joint response on the first page of this document.

More specifically, we think it is important to stress out that an important conclusion of our work, which distinguish it from previous studies on solid tumors, is that polarity defect in the blood cell could be induced by the transformation of either the stromal cell or the blood cell.

Despite its descriptive nature, the establishment of a clear correlation between leukemia and polarity defects in blasts unveils novel avenues for the study of leukemia, akin to the seminal early descriptions of the role of cell polarity in solid tumors. Independently of leukemia, our work also suggests new hypotheses regarding the role of cell adhesion and polarity in the regulation of HSC proliferation and differentiation. As such, we believe it represents a significant advance in our understanding of the HSC regulation and leukemia progression, and therefore deserves publication in a broad readership journal such as EMBO Reports.

2. Does leukemic and normal cell polarization (or no polarization) towards or away from stromal cells as shown in figure 2 alter the fate of the daughter cells post cell division? Thus, would there be a **shift in symmetric vs. asymmetric cell division**?

This is a great suggestion. We have recently shown that the polarization of HSC affects the orientation of their mitotic spindle and the symmetry of their division via the CXCR4/CXCL12 pathway (Candelas et al., 2023). In parallel, it has been shown that the misorientation of the mitotic spindle in AML blasts affects their differentiation and contributes to the progression of myeloid leukemia (Baja et al., Blood, 2017). Thus, it is indeed likely that the defective polarization we observed may contribute to leukemia progression via its effect on mitotic spindle orientation. However, the methodological challenges involved in characterizing the specific fate of daughter cells produced by HSCs, polarized or not, through their contact with stromal cells, make these experiments particularly difficult to perform.

3. Do magnupodia enrich for any known markers of cell polarity? Do the HSPCs and AML cells co-cultured with stromal cells display **polarized expression of Cdc42 and/or Numb?**

Our previous work showed the enrichment of myosin and ezrin in the magnupodium and the accumulation of CD44 on the opposite pole (Bessy et al., J Cell Biol, 2021). To address reviewer's comment more specifically, we studied the localization of CD133/prominin1, Numb and Cdc42.

3a. CD133/Prominin

The formation of cellular protrusions known as magnupodia has been associated with the polarization of specific proteins, including the glycoprotein CD133 (Prominin-1) (Freund et al. 2006). Conditions such as hypoxia, mitochondrial dysfunction, or mitochondrial DNA depletion have been shown to

reversibly upregulate CD133 expression (Griguer et al. 2008). Freund et al. further demonstrated a selective concentration of CD133 within magnupodia.

To address reviewer's comment, we investigated the distribution of CD133; however, no specific pattern was observed (see right panels below showing CD133 distribution within a polarized HSPC interacting with an osteoblast). A significant limitation in studying CD133 lies in the specificity of available antibodies (130-090-422 Miltenyi Biotech), which detect only its glycosylated forms, complicating immunolabeling and leading to reproducibility challenges. Developing novel reagents capable of recognizing splice variants and post-transcriptionally modified forms of CD133 would significantly enhance future investigations. Exploring the distribution of CD133 in AML blasts compared to healthy HSPCs during cell division remains a promising area for further elucidating its role in hematopoietic and leukemic biology

3.b. Numb

The cell fate determinant Numb has been shown to segregate asymmetrically during cell division, with ectopic expression promoting differentiation in AML cells, while its inhibition leads to an increase in symmetric divisions (Zimdahl et al. 2014). Investigating the segregation and inheritance of Numb in progeny cells derived from AML blasts, in comparison to healthy HSPCs, could provide insights into the mechanisms underlying the imbalance between asymmetric and symmetric divisions observed in leukemia. However, immunofluorescence analysis of Numb distribution in both polarized and non-polarized healthy HSPCs did not yield conclusive results, as the antibodies we tested (**ab4147, Abcam**) revealed a concentration of signal at the centrosome rather than the expected localization at the cell cortex.

3.c. Cdc42

Polarity has been shown to be disrupted in AML cells, with Cdc42 identified as a key determinant of cellular polarity. Evidence suggests a causal relationship between the upregulation of the small Rho GTPase Cdc42 and the decline in polarity observed during the aging of HSCs (Florian et al. 2012)

We followed reviewer's suggestion and analyzed the localization of Cdc42 in both healthy and leukemic hematopoietic cells in contact with osteoblast (left). In HSPC, although the signal appeared uniformly distributed in the cytoplasm, it ended up being more present on the side toward the contact site due to cell deformation. In LSC, the round cell shape came with more even distribution of Cdc42.

To further investigate the possibility that cell deformation and the polarization of internal organization was associated with Cdc42 enrichment, we studied Cdc42 in healthy CD34+ HSPCs or LSC plated on a glass surface coated with CXCL12, since we showed previously that these conditions lead to the polarization of isolated HSPC (Bessy et al., JCB, 2021). We found that in both healthy and transformed cells Cdc42 was asymmetric because it concentrated around the centrosome. The main difference was that in polarized healthy HSPC the asymmetry was oriented toward the contact with CXCL12 whereas in LSC, which were not polarized, it was randomly orientated.

Our transcriptomic analyses showed large variations of the level of expression of Cdc42 between samples, and no clear difference between healthy and leukemic cells. Our immunostainings show that Cdc42 was intrinsically asymmetrically distributed in all cells but orientated in polarized healthy HSC only. These data suggest that Cdc42 concentration and localization respond to but did not drive cell polarization.

- While the methods indicate that single HSPCs/AML cells were seeded along with single MSCs, the images shown in Figs. 1-3 seem to indicate that more than one hematopoietic cell was present in many cases. This is likely due to the technical challenges associated with single cell seeding. However, this does raise the question whether **stromal cell contact with two or more hematopoietic cells alter polarization as compared to one-one cell contact?**

We agree that multiple contacts between hematopoietic cells and a single stromal cell could affect the concentration of receptors on its surface and thus the polarization of hematopoietic cells. We compared the polarity index of healthy CD34+ HSPC interacting with a single osteoblast depending on their number per well. We found that it did not have any significant impact on the positioning of their centrosome toward the contact sites (polarity index), nor did it affect their ability to form magnupodia.

Polarization index of cord blood CD34+ HSPC as a function of the number of HSPCs in contact with a single osteoblast (n_{singlet HSPC} = 55, n_{doublet HSPCs} = 42; n_{triplet HSPCs} = 12)

- It would be helpful to include **normal differentiated cells** as controls to determine if lack of polarization is specific to the leukemic state and not simply a readout of progenitor/mature cells behavior, especially since the LSC content in all AML samples used is fairly low. In addition, data should be included from AML samples enriched for CD34+ cells as is done for normal bone marrow.

This is an interesting remark. In our previous study we characterized the effect of cell differentiation on cell polarization and found that differentiated cells could not polarize toward niche stromal cells (Bessy et al. 2021). In co-culture with osteoblasts, both the CD34⁺CD38^{low} hematopoietic stem cell-enriched population and the CD34⁺CD38⁺CD33⁺ CMPs exhibited polarization, whereas neither monocytes (CD14⁺) nor primary lymphocytes (CD3⁺) displayed a cellular phenotype indicative of polarization. This could indeed suggest that the defective polarity in leukemic blast could be associated to their differentiation and loss of stemness while conserving their high proliferative capacities.

To further address reviewer's comment about LSC, we searched in our AML patient bank database and identified a sample with an enrichment of CD34⁺ cells (patient° 10 with 32% LSCs). To enrich the LSC fraction from this bone marrow sample, magnetic purification was performed to isolate CD34⁺, providing a less invasive method of sorting (compared to classical FACS) subsequently followed by a pre-incubation with osteoblasts in the microwells. While a small proportion of cells displayed magnupodia, the majority exhibited a rounded blast-like morphology, with random centrosome positioning relative to the stromal cell contact. **These data have been included in the new Figure 2B.**

Minor Comments:

1. The references have several discrepancies that need to be corrected.

We thank the reviewer for paying attention to this mistake. We corrected it.

Referee #3

The manuscript "AML patient blasts exhibit polarization defect upon interaction with bone marrow stromal cells" by Saadallah K. et al, investigates the capacity of leukemic stem cell (LSC) to polarize upon contact with the niche cells.

The investigation of LSC polarity upon contact with niche cells is novel and interesting. However, the data included in the current version of the manuscript requires some important clarifications and/or some additional experiments are needed. The major points to be addressed are:

- 1) The **role of polarity in AML** has been associated with AML development and survival. Cell polarity is important for cell migration/motility but in the context of AML it is somehow not so critical as cell proliferation or AML survival/development. Can the authors investigate **leukemic cell proliferation/survival alongside polarization?**

This is a great suggestion. However, the issue is that it is clear that blasts won't polarize (as shown by this study) and will proliferate (as it is well known) but this won't demonstrate that both are causally linked. To challenge such a causal relationship it is necessary to control blast (or healthy HSPC) polarization and see whether this affects their proliferation. To do so, one need to identify the molecular pathway regulating cell polarization and modulate it in order to reveal its consequences on cell proliferation. This represents a lot of work. The other reviewers made similar comments and the reviewer is encouraged to read the joint response on the first page of this document.

To start investigating the molecular mechanism involved in HSC polarization, we performed a transcriptomic analysis of all the patient samples we used in the study. Here is the heatmap of the main adhesion and polarity genes displaying differential expression between healthy CD34+ cord blood HSPC (CB) and AML patient samples (blast and LSC). As expected, CB exhibited lower expression of proliferation-related genes (CCNE1, CCND1, CCNB1, CDKN2C, E2F1, MTK67 and PCNA) as compared to leukemic blasts and more particularly to LSCs. Genes related to cell adhesion, cytoskeleton and polarity genes displayed less obvious changes. Some variations such as tubulin and stathmin (microtubule stabilization), or Prominins and Numb (cortical polarity markers), may indicate some genuine differences but these would need more samples to be confirmed since the statistical analyses of the current data set did not validate their significance.

2) Another important aspect is related with the overall study design, where the authors compare CD34⁺ HSPCs from healthy donor with unenriched mononuclear cells from AML patients (with very high blast counts). The leukemic blasts used in the study must be better characterized by **flow cytometry**. Ideally it would be **more relevant to compare LSC vs HSPCs**.

We performed additional experiments in order to provide a more detailed characterization of the AML blasts used in this study through flow cytometry analysis. It is now provided in the new supplementary figure S1.

Furthermore, to better characterize the polarization capacities of LSC, we analyzed cells from an additional patient in our data base. The sample was characterized by a significant presence of LSCs, defined by the phenotype CD34⁺ CD38⁻ CD13⁺ CD33⁺ CD7⁺ partial CD123⁺ partial CD97/CLL1/TIM3⁺ CD45RA⁻ (Supplementary Figure S1 Table 3). Hematopoietic cells from AML patients were analyzed using freshly obtained bone marrow (BM) samples. A three-tube panel was employed to assess the Leukemia Aberrant Immunophenotype (LAIP) and Different-from-Normal (Dfn) patterns, identify CD34⁺CD38⁻ LSCs, and evaluate monocytic blast differentiation, as outlined by a recent work conducted by the Hematology Department of Saint Louis hospital (ASH abstract 2024 #226 (Plesa et al. 2024)). We found that, as all blasts, they were not capable of polarizing toward their contact site with stromal cells. **These data have been included in the new Figure 2B.**

3) Along the same line, also the healthy and leukemic niche should be better characterized by providing some flow cytometry analysis of the niche cells used for the polarity assay.

We apologize for this lack of characterization. The primary BM niche cells, derived from either healthy donors (HD donor°6 and donor°12) or AML patients (patient°4 and patient°6) have now been better characterized using flow cytometry and are presented in the **new supplementary figure (Figure S1 5a.b)**.

4) It is likely that each patient has different frequency and composition of LSC and niche cells. How do the authors take this aspect into consideration?

This is a fair remark. The considerable clinical heterogeneity observed in AML at both the genetic and phenotypic levels is a challenge. To address the significant variability in blast and/or LSC frequencies among patients, we selected individuals within the same age range with *de novo* AML, harboring the most frequent driver mutations and ensuring that the cells had not been subjected to any prior treatment. We chose samples with the highest percentage of blasts. Subsequent analysis revealed that the AML cells were predominantly blasts, with very low or undetectable levels of LSCs (around 1% of LSCs).

5) The characteristics (number, age, sex etc) of the healthy donor cohort are not provided.

For the healthy donor cohort, age and gender are the only available clinical data that we can provide. The limited availability of bone marrow samples from femoral fragments, coupled with the scarcity of CD34 positive sorted cells, has constrained the extent of further analysis.

6) In figure 4 the same picture is used in panel Ad and Bb with different labels.

We apologize for this oversight and thank the reviewer for this careful reviewing of our work. Another image has been provided to replace the image that was used to illustrate both the endosteal compartment and the MOLM-1 AML cell line within the endosteal compartment.

7) The last sentence of the abstract is an overstatement. there is no data in the manuscript supporting this claim because there is no analysis of leukemic progression nor of the causality between defective polarization of AML blasts and leukemic progression.

We acknowledge that it is inappropriate to mention potential future treatments since the causal relationship between leukemia progression and cell polarization defects is not established. The last sentence of the abstract has been deleted.

Dear Dr. They,

Thank you for the submission of your revised manuscript to our editorial offices. I have now received the reports from the three referees that were asked to re-evaluate the study, you will find below. As you will see, the referees point out some limitations of the study, but now support its publication in EMBO reports.

Before I can proceed with formal acceptance, I have these editorial requests I ask you to address in a final revised manuscript:

- Please chose one of the two titles:

"AML patient blasts exhibit polarization defects upon interaction with bone marrow stromal cells" or "AML patient blasts exhibit a polarization defect upon interaction with bone marrow stromal cells".

- Please provide an abstract with not more than 175 words, written in present tense.

- We plan to publish your manuscript as Report, as there are only 5 main figures and 3 EV figures. For a Scientific Report we require that results and discussion sections are combined in a single chapter called "Results & Discussion". Please do this for your manuscript. For more details please refer to our guide to authors:

<http://www.embopress.org/page/journal/14693178/authorguide#researcharticleguide>

- We now use CRediT to specify the contributions of each author in the journal submission system. CRediT replaces the author contribution section. Please use the free text box to provide more detailed descriptions and do NOT provide your final manuscript text file with an author contributions section. See also our guide to authors:

<https://www.embopress.org/page/journal/14693178/authorguide#authorshipguidelines>

- We request that primary datasets produced in a study (e.g. RNA-seq, ChIP-seq, structural and array data) are deposited in an appropriate public database. If no primary datasets have been deposited, please also state this in a dedicated section (e.g. 'No primary datasets have been generated and deposited').

- Please reduce the number of keywords to five and order the manuscript sections like this, using these names:

Title page - Abstract - Keywords - Introduction - Results & Discussion - Methods - Data availability section - Acknowledgements (including the funding information) - Disclosure and Competing Interests Statement - References - Figure legends - Expanded View Figure legends

- Please use our reference format:

DOIs should only be used for preprints and datasets that have not been published yet.

- Please make sure that all the funding information is also entered into the online submission system and that it is complete and similar to the one in the acknowledgement section of the manuscript text file. Presently, information related to 'the Bettencourt-Schueller Foundation, the Emergence program of the Ville de Paris and the Schlumberger Foundation for Education and Research. The Integrated Cancer Research Center, "SiRIC InSiTu: Insights into Cancer: From Inflammation to Tumor," funded this study, under grant number INCa-DGOS-INSERM-ITMO Cancer_18008; doctoral school contract provided by the University of Paris Cité, as well as from The National Center for Precision Medicine in Leukemia (THEMA) of Saint Louis Hospital (Paris, France) and Bristol Myers Squibb (BMS)' is missing in the submission system. All these funders need to be entered via the 'More Funders' option (the comments box should not be used).

- Please check again that the number "n" for how many independent experiments were performed, their nature (biological versus technical replicates), the bars and error bars (e.g. SEM, SD) and the test used to calculate p-values is indicated in the respective figure legends. Please also check that all the p-values are explained in the legend, and that these fit to those shown in the figure. Please provide statistical testing where applicable. Please avoid the phrase 'independent experiment', but clearly state if these were biological or technical replicates. Please also indicate (e.g. with n.s.) if testing was performed, but the differences are not significant. In case n=2, please show the data as separate datapoints without error bars and statistics. See also:

<http://www.embopress.org/page/journal/14693178/authorguide#statisticalanalysis>

If n<5, please show single datapoints for diagrams. Moreover:

- Please note that the legends for figures 2 is not provided in the sequential manner (legend for figure 2C is provided before legend of figure 2B). This needs to be rectified.

- Please note that the figure 2D is missing in the manuscript, however legend for the same is provided. This needs to be rectified.

- Please note that the exact p values are not provided in the legends of figures 1E, 2B, 3B, 4D, 5B, C; EV3 C, D.

- Please make sure that all figure panels (main and EV figures) are called out separately and sequentially. Presently, a panel 2D is called out but the panel is missing in the figure (although there is a legend for 2D); a figure S1 is called out (could this refer to Figure EV1?) and a Table 4 called out but missing. Please check.
- The movie file is missing a legend and needs to be named "Movie EV1". The legend should be provided in a readme.txt file and then be ZIPed together with the movie file and uploaded as Movie EV1.
- All Materials and Methods need to be described in the main text using our 'Structured Methods' format, which is required for all research articles. According to this format, the Methods section should include a Reagents and Tools Table (listing key reagents, experimental models, software, and relevant equipment and including their sources and relevant identifiers), uploaded as separate file, and a Methods section in which we encourage the authors to describe their methods using a step-by-step protocol format with bullet points, to facilitate the adoption of the methodologies across labs. More information on how to adhere to this format as well as downloadable templates (.doc) for the Reagents and Tools Table can be found in our author guidelines (section 'Structured Methods'):

Please add the antibody table to the Reagents & Tools table and remove it from the Methods section.

- Please add scale bars of similar style and thickness to microscopic images, using clearly visible black or white bars (depending on the background). Please place these in the lower right corner of the images themselves. Please do not write on or near the bars in the image but define the size in the respective figure legend. Presently, all scale bars have text nearby.
- Please provide the source data as requested. Please upload the SD as one folder per figure, grouping together separate excel files for all panels for one figure (and ZIPed together) and as one folder for the EV figures. If the source data has been deposited in an archive, please provide the full link in the Data Availability Section (see above).

In addition, I would need from you uploaded separately:

Best,

Referee #1:

The authors have addressed most of our concerns.

Referee #2:

The authors have adequately addressed most of my major concerns. However, it would be helpful to clarify the new results in figure 2b. The X axis labeling suggests that the last sample (pt. 10) has data from both LSCs and blasts. Unfortunately, it is not clear from the graph if the LSCs and blasts from Pt. 10 have similar polarization index. It may be helpful to color the LSCs differently from the blasts to clearly demarcate them.

Overall, this is an interesting piece of work on a relatively under-studied aspect of leukemia biology. This study lays an important foundation for future mechanistic and functional studies aimed at defining the role of leukemic cell polarization on disease progression.

Referee #3:

The authors have addressed all my requests beside the first one. To justify the inability to respond to the first question (that was similarly ask by all the referees), they provided a preliminary dataset to share some of the molecular markers that were identified to differ between the different samples and possibly are involved in polarization. However, inter-patient variability didn't allow to conclude on a definitive target and based on this the authors concluded that the link between defective polarity and proliferation can't be conclusively established. Therefore, they are not able to provide any data about leukemia proliferation alongside polarization.

While this referee understands the authors' perspective and appreciates the transparency in sharing the preliminary data and the difficulties to move forward with the request in the short time allowed for the revision, the absence of this data limits the implications of the findings and therefore the overall significance of the study.

The rest of the data have been revised and are now appropriate to move forward with publication.

All editorial and formatting issues were resolved by the authors.

Dr. Manuel They
CEA, Hopital Saint-Louis, Paris
1 Avenue Claude Vellefaux
Paris 75010
France

Dear Dr. They,

I am very pleased to accept your manuscript for publication in the next available issue of EMBO reports. Thank you for your contribution to our journal.

Yours sincerely,
